# Itaconate ameliorates autoimmunity by modulating T cell imbalance via metabolic and epigenetic reprogramming

Kuniyuki Aso [1], Michihito Kono [1] ✉, Masatoshi Kanda[2], Yuki Kudo[1], Kodai Sakiyama [1], Ryo Hisada[1], Kohei Karino[1], Yusho Ueda[1], Daigo Nakazawa [1], Yuichiro Fujieda[1], Masaru Kato [1], Olga Amengual[1] & Tatsuya Atsumi [1]

Dysregulation of Th17 and Treg cells contributes to the pathophysiology of many autoimmune diseases. Herein, we show that itaconate, an immunomodulatory metabolite, inhibits Th17 cell differentiation and promotes Treg cell differentiation by orchestrating metabolic and epigenetic reprogramming. Mechanistically, itaconate suppresses glycolysis and oxidative phosphorylation in Th17- and Treg-polarizing T cells. Following treatment with itaconate, the S-adenosyl-L-methionine/S-adenosylhomocysteine ratio and 2-hydroxyglutarate levels are decreased by inhibiting the synthetic enzyme activities in Th17 and Treg cells, respectively. Consequently, these metabolic changes are associated with altered chromatin accessibility of essential transcription factors and key gene expression in Th17 and Treg cell differentiation, including decreased RORγt binding at the *Il17a* promoter. The adoptive transfer of itaconate-treated Th17-polarizing T cells ameliorates experimental autoimmune encephalomyelitis. These results indicate that itaconate is a crucial metabolic regulator for Th17/Treg cell balance and could be a potential therapeutic agent for autoimmune diseases.

Autoimmune diseases are characterized by the loss of self-tolerance and systemic inflammation that targets vital organs[1]. Immunosuppressive agents play a central role in the treatment of autoimmune diseases. Conventional drugs, including corticosteroids, are broadacting and increase the risk of severe infection, which is a leading cause of death[1,2]. Although more targeted drugs against distinct immune cells or cytokines have been developed, the balance between their efficacy and side effects is still challenging[2]. Considering the unsatisfactory remission rate in the treatment with these drugs[3], more specific treatments targeting the pathogenic mechanisms underlying autoimmune diseases are required.

Upon antigen stimulation in the presence of unique cytokine signals and microenvironment, distinct T cell subsets differentiate from naive CD4+ T cells[4]. Although T helper 17 (Th17) and regulatory T (Treg) cells require a common tumor growth factor (TGF)-β signal for

their differentiation[5], these cells fulfill opposite functions. Th17 cells play a pathogenic role in several autoimmune diseases, while Treg cells maintain immune homeostasis and inhibit autoimmunity[6]. Dysregulation of Th17 and Treg cells contributes to the pathophysiology of many autoimmune diseases[7], including multiple sclerosis (MS)[8], systemic lupus erythematosus[9], and rheumatoid arthritis[10]. However, therapy targeting Th17/Treg cell imbalance has not been established in clinical settings[11]. The balance between these T cell subsets depends on cellular metabolism, which alters cellular epigenetics and transcription, and modulates their effector functions[4]. Effector T cells, such as T helper 1 (Th1) and Th17 cells, depend primarily on glycolysis, whereas Treg cells utilize oxidative phosphorylation (OXPHOS) and fatty acid oxidation (FAO) for survival, differentiation, and effector function. Failure to induce appropriate metabolic pathways impairs differentiation and effector function in activating CD4+ Th cells from the

---

[1]Department of Rheumatology, Endocrinology and Nephrology, Faculty of Medicine and Graduate School of Medicine, Hokkaido University, Sapporo, Japan.
[2]Department of Rheumatology and Clinical Immunology, Sapporo Medical University, Sapporo, Japan. ✉e-mail: m-kono@med.hokudai.ac.jp

naïve T cells[4]. Pharmacological inhibition of pyruvate kinase muscle isozyme 2, which is the final rate-limiting enzyme in glycolysis, reduces glycolysis and limited Th1 and Th17 cell differentiation in vitro[12]. Furthermore, recent studies have shown that some metabolites regulate Th17 and Treg cell differentiation. Reduced universal methyl donor S-adenosyl-L-methionine (SAM) levels, following restriction of methionine metabolism, decreases histone methylation at the *Il17a* promoter and restricts Th17 cell differentiation[13]. During Treg cell differentiation, the reduction of 2-hydroxyglutarate (2-HG) levels, following inhibition of glutaminolysis, induced demethylation of *Foxp3* and promoted Treg cell differentiation[14]. Previous results have suggested that T cell function and differentiation are determined by the changes of crucial "metabolites" due to the altered balance of the distinct metabolic pathways.

Itaconate (ITA) is an endogenous metabolite derived from the mitochondrial tricarboxylic acid (TCA) cycle. In recent years, ITA has gained attention owing to its anti-inflammatory, antiviral, and antimicrobial effects[15–17]. ITA inhibits the production of proinflammatory cytokines, including interleukin-1β (IL-1β) and IL-6, through various mechanisms in macrophages[18–20]. ITA has an antimicrobial effect with direct inhibition of the bacterial isocitrate lyase[21] and potently inhibits viral replication and excessive inflammatory host responses to human pathogenic viruses, including severe acute respiratory syndrome coronavirus 2[22]. Furthermore, ITA inhibits key glycolytic enzymes and impairs glycolysis in macrophages[20]. However, its immunomodulatory role in T cell differentiation and function has not been elucidated thoroughly.

Here, in this study, we aim to identify the role of ITA in regulating T cell differentiation and its potential as a candidate to treat T cell-

mediated autoimmune diseases. Consequently, we identify ITA supplementation inhibits Th17 cell differentiation and promotes Treg cell differentiation through metabolic and epigenetic reprogramming. Furthermore, we demonstrate that ITA ameliorates the disease activity of T cell-driven autoimmune disorder model.

## Results

### Itaconate modulates Th17 and Treg cell differentiation

To determine if ITA influences T cell differentiation in vitro in naive CD4[+] T cells isolated from the spleen of C57BL/6 J mice were cultured under Th1, Th2, Th17, and Treg-polarizing conditions, with or without ITA (Supplementary Fig. 1a). ITA inhibited Th17 and promoted Treg cell differentiation in a dose-dependent manner (Fig. 1a, b), without affecting their cell viability, absolute number, and proliferation (Supplementary Fig. 1b−d). In contrast, ITA did not impact the differentiation of Th1 and Th2 cells. (Fig. 1a, b). ITA reduced the expression of Th17-related genes, including *Il17a* and *Il17f*, while ITA enhanced the expression of *Rorc*, which encodes a master Th17 lineage transcription factor RORγt in Th17-polarizing T cells (Fig. 1c). Additionally, ITA enhanced the expression of *Foxp3* in Treg-polarizing T cells (Fig. 1c). Protein levels of RORγt were also elevated in Th17-polarizing T cells (Fig. 1d). In pathogenic Th17 cells induced by IL-6, IL-1β, and IL-23, ITA inhibited the production of IL-17 and granulocyte-macrophage colony-stimulating factor (GM-CSF) (Supplementary Fig. 1e). These results indicated that ITA regulated Th17 and Treg cell differentiation.

ITA is a metabolite produced by macrophages in which *Irg1* is highly expressed. In contrast, our study demonstrated no upregulation of *Irg1* in T cells (Supplementary Fig. 1f). Moreover, a previous study has shown that ITA could be released from activated macrophages at

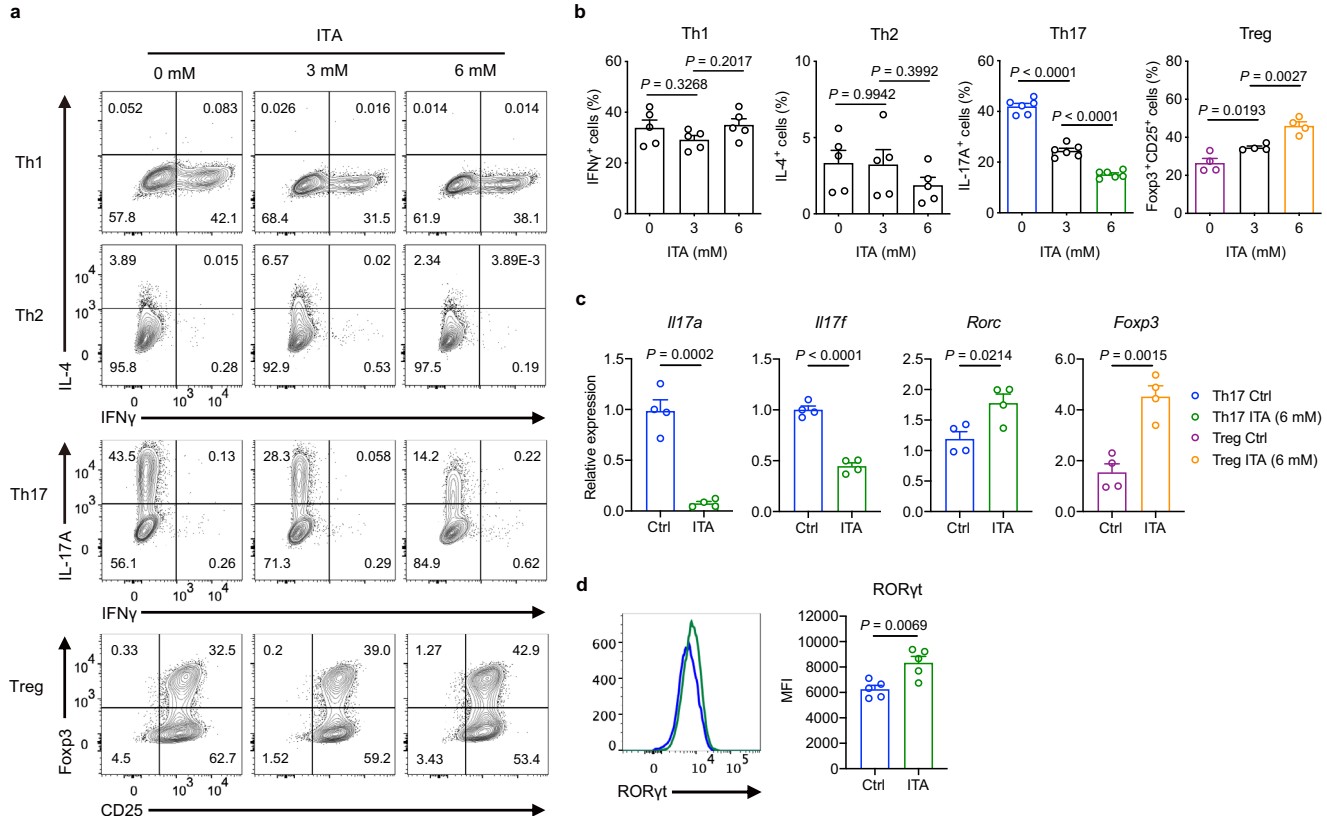

**Fig. 1 | Itaconate inhibits Th17 differentiation and enhances Treg differentiation.** Representative flow plots (**a**) and cumulative data (**b**) of the differentiation of murine naive CD4[+] T cells from wild-type B6 mice activated under Th1, Th2, Th17, and Treg cell conditions in the presence or absence of itaconate (ITA; 0, 3, and 6 mM) after 3 days culture (Th1 and Th2, *n* = 5; Th17, *n* = 6; Treg, *n* = 4). **c** Expression of Th17- and Treg-related genes in the presence or absence of ITA (0 and 6 mM)

(*n* = 4, each condition). **d** Mean fluorescence intensity (MFI) of RORγt expression under Th17-polarizing condition in the presence or absence of ITA (6 mM) (*n* = 5). *P* values are calculated using one-way ANOVA with Bonferroni post hoc test for (**b**) and two-tailed unpaired Student's *t*-test for (**c**, **d**). Data are representative of mean ± s.e.m. Source data are provided as a Source Data file.

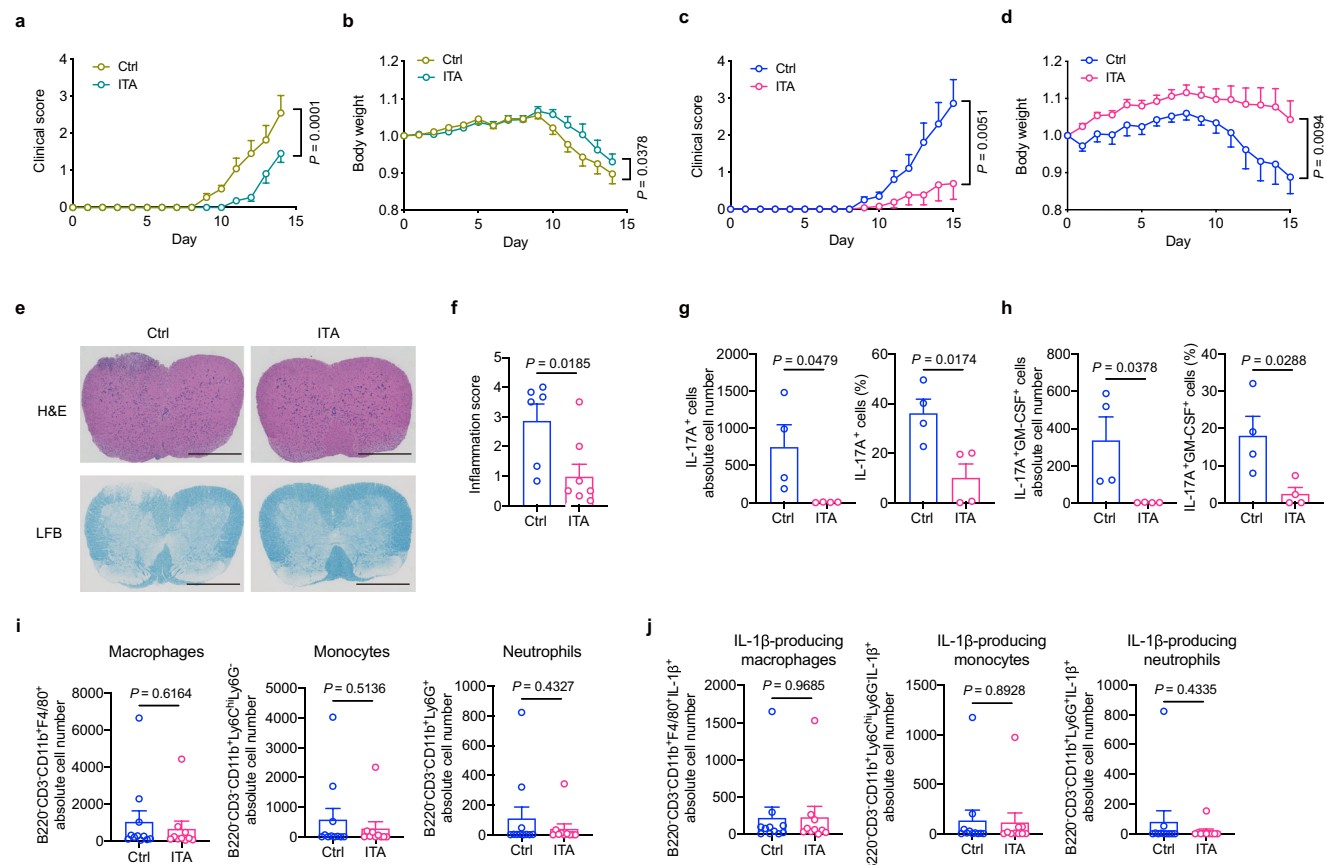

**Fig. 2 | Itaconate ameliorates experimental autoimmune encephalomyelitis.**
**a**, **b** C57BL/6J mice were immunized with myelin oligodendrocyte glycoprotein (MOG)$_{35-55}$ and complete Freund's adjuvant (CFA). Mice were intraperitoneally injected with 50 mg kg$^{-1}$ itaconate (ITA) every other day from day 0 to day 14. Clinical scores (**a**) and body weight (**b**) in EAE mice treated with PBS (Ctrl, $n = 11$) or ITA ($n = 11$). **c**–**j** For adoptive transfer EAE (tEAE), pathogenic Th17-polarizing CD4$^+$ T cells from 2D2 mice were cultured with or without ITA ex vivo for 3 days. Then, the harvested cells were transferred to recipient *Rag1*-deficient mice intravenously. Clinical scores (**c**) and body weight (**d**) of control (Ctrl, $n = 10$) or ITA ($n = 13$) recipient mice in tEAE models. **e** Representative histology of spinal cord stained with hematoxylin and eosin (H&E) and luxol fast blue (LFB). Scale bar, 500 μm. **f** Inflammation scores of spinal cords are shown (Ctrl, $n = 6$; ITA, $n = 8$, biologically

independent samples). Absolute numbers (left) and frequency (right) of IL-17A$^+$ (**g**) and IL-17A$^+$GM-CSF$^+$ (**h**) CD4$^+$ T cells in the spinal cord of recipient mice, as assessed using flow cytometry ($n = 4$ in each condition, biologically independent samples). **i** Absolute number of macrophages, Ly6C$^{hi}$ monocytes, and neutrophils in the spinal cord of *Rag1*-deficient mice on day 14 after induction of EAE (Ctrl, $n = 11$; ITA, $n = 10$, biologically independent samples). **j** Absolute number of IL-1β-producing cells in three cell types as described in (**i**) (Ctrl, $n = 11$; ITA, $n = 10$, biologically independent samples). Data of **e** are representative of four independent experiments with similar results. *P* values are calculated using two-way ANOVA for (**a**–**d**) and two-tailed unpaired Student's *t*-test for (**f**–**j**). Data are representative of mean ± s.e.m. Source data are provided as a Source Data file.

an extracellular concentration of 1–5 μM[23]. However, here, we demonstrated that the effect of ITA on Th17 and Treg differentiation was not statistically significant at a lower concentration than 1 mM (Supplementary Fig. 1g, h). Altogether, these results suggested that T cell differentiation was not affected by macrophage-derived ITA in vivo.

**Itaconate ameliorated the EAE model**
To investigate the potential of ITA as a candidate to treat autoimmune diseases, we used an experimental autoimmune encephalomyelitis (EAE) model in vivo. EAE mice were intraperitoneally injected with 50 mg kg$^{-1}$ ITA every other day from day 0 to day 14 post-immunization with myelin oligodendrocyte glycoprotein (MOG)$_{35-55}$ and complete Freund's adjuvant (CFA). ITA treatment significantly reduced the clinical scores and loss of body weight in EAE mice compared to PBS treatment (Fig. 2a, b). These results indicated the potential of ITA to reduce the disease activity of EAE.

The MOG-induced EAE model could not strictly exclude the possible effects of ITA on cell types other than T cells, such as the anti-inflammatory effect on macrophages. To further investigate the role of ITA on in vivo functionality of Th17, pathogenic Th17-polarizing CD4$^+$

T cells from 2D2 mice were cultured ex vivo for 3 days following adoptive transfer into *Rag1*-deficient mice (Supplementary Fig. 2a). ITA treatment reduced the percentage of IL-17A and GM-CSF-producing CD4$^+$ T cells in ex vivo culture (Supplementary Fig. 2b). Clinical scores and loss of body weight were significantly reduced in *Rag1*$^{-/-}$ mice administered ITA-treated cells compared to those in mice administered with control cells (Fig. 2c, d). Histological sections of spinal cords showed significantly reduced cell infiltration and demyelination in mice administered ITA-treated cells (Fig. 2e, f). Furthermore, mice administered with ITA-treated Th17 cells showed a significantly reduced number and percentage of IL-17A-producing CD4$^+$ T cells, as well as IL-17A and GM-CSF-producing CD4$^+$ T cells, in the spinal cord 15 days after EAE induction compared to that in control mice (Fig. 2g, h). We next evaluated the infiltrating macrophages, inflammatory Ly6C$^{hi}$ monocytes, and neutrophils, which drive T cells that mediate pathology in EAE[24], in the spinal cord of the adoptive transfer EAE (tEAE) model. The absolute number of these infiltrated cells or IL-1β-producing cells in recipient mice administered ITA-treated Th17 cells did not differ significantly from that in their counterparts (Supplementary Fig. 2e and Fig. 2i, j). These data suggest that in vivo induction of neutrophils, inflammatory monocytes, and macrophages

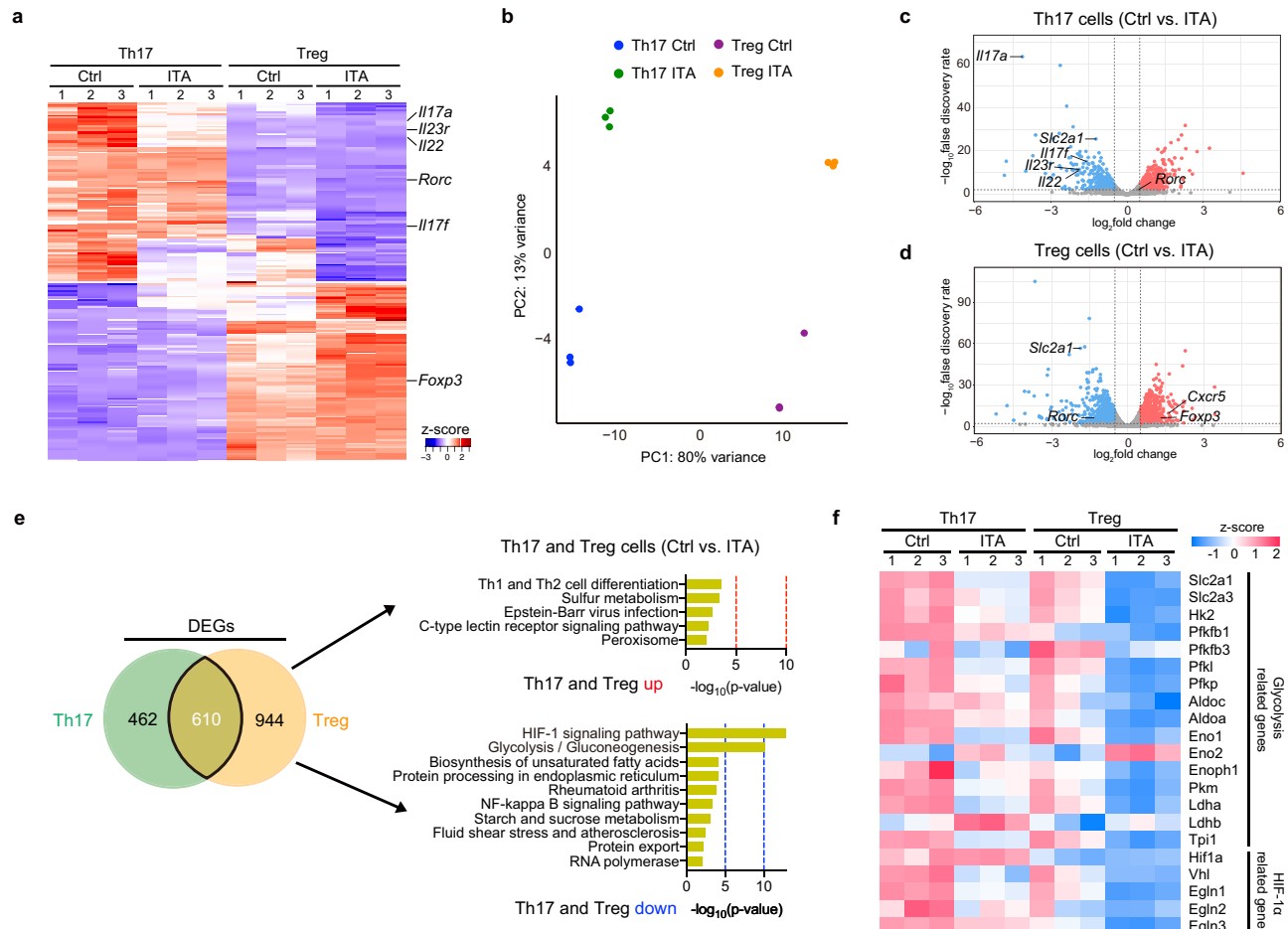

**Fig. 3 | Itaconate decreases glycolysis-related genes under Th17- and Treg-polarizing conditions. a–f** Th17- and Treg-polarizing T cells from B6 mice in the presence or absence of itaconate (ITA) after 2 days of culture were subjected to RNA sequencing ($n = 3$ in each condition, independent experiments). Heatmap (**a**) shows the top 100 and bottom 100 modified genes from RNA-seq in Th17- (left) and Treg- (right) polarizing T cells in the presence or absence of ITA. Principle component analysis of global gene expression from RNA-seq in four T cell populations (**b**). Volcano plot of differential gene expression in ITA-treated Th17- (**c**) or Treg- (**d**) polarizing T cells compared to Control (Ctrl). Venn diagram (**e**) displays the overlapping DEGs between ITA-treated Th17- and Treg-polarizing T cells. The top gene ontology pathways in the overlapping group are shown. *P* value for (**e**) indicates gene enrichment analysis test implemented in Metascape without adjustment for multiple comparisons. Heatmap (**f**) shows relative expression (z-score) of glycolysis and HIF-1α-related genes according to RNA-seq. Source data are provided as a Source Data file.

is unlikely to be the main pathogenicity of the tEAE attenuation by ITA-treated Th17 cells. Additionally, the intraperitoneal injection of ITA to *Rag1*-deficient recipient mice following the adoptive transfer of Th17-polarizing CD4+ T cells from 2D2 mice significantly attenuated the severity of the adoptive tEAE model (Supplementary Fig. 2c, d). Collectively, these findings suggest that ITA is a potential therapeutic agent to treat T cell-driven autoimmune disorders.

**Itaconate inhibits glycolysis and OXPHOS under Th17- and Treg-polarizing conditions**

To understand the mechanisms underlying ITA-dependent modulation of T cell differentiation, we performed RNA sequencing (RNA-seq) using Th17- and Treg-polarizing T cells with or without ITA treatment (Fig. 3a, b). A total of 1072 differentially expressed genes (DEGs) were identified between ITA-treated and control Th17-polarizing T cells (Fig. 3c). In ITA-treated Treg-polarizing T cells, 1554 genes were identified as DEGs, of which 610 DEGs were identified in Th17-polarizing T cells as well (Fig. 3d). Kyoto Encyclopedia of Genes and Genomes (KEGG) pathway analysis identified that some metabolic pathways were enriched in Th17- and Treg-polarizing T cells (Supplementary Fig. 3a, b). Among Th17 signature genes[25], the relative expression of genes that encode the effector cytokine IL-17A (*Il17a*) and F (*Il17f*) and

signal transducer and activator of transcription 3 (*Stat3*) were downregulated by ITA under Th17 condition compared to that in control (Fig. 3c and Supplementary Fig. 3c). Interestingly, the relative expression of gene encoding important transcription factor for Th17 polarization, including RORγt (*Rorc*) was not downregulated by ITA (Supplementary Fig. 3c). KEGG pathway analysis of the overlapped DEGs between ITA-treated and control T cells under Th17 and Treg conditions revealed significant enrichment of 'glycolysis' and 'HIF-1 signaling pathway' (Fig. 3e). The results showed that several glycolysis-related genes, including the glucose transporter *Slc2a1* (encoding Glut-1), were downregulated in ITA-treated T cells under Th17- and Treg-polarizing conditions (Fig. 3f). Glut-1 regulates glucose uptake and glycolysis upon naive CD4+ T cell activation and differentiation[26]. These results indicated that ITA broadly inhibited glycolysis-related gene expression.

Hypoxia-inducible factor 1 alpha (HIF-1α) is an important regulator of glycolysis[4]. HIF-1α-deficient T cells demonstrate suppressed glycolysis, resulting in reduced Th17 cell differentiation and increased Treg cell differentiation[27]. ITA modestly increased *Hif1a* gene expression (Fig. 3f) and HIF-1α protein levels under Th17- and Treg-polarizing conditions (Fig. 4a, b). We next performed glycolytic rate assays using an extracellular flux

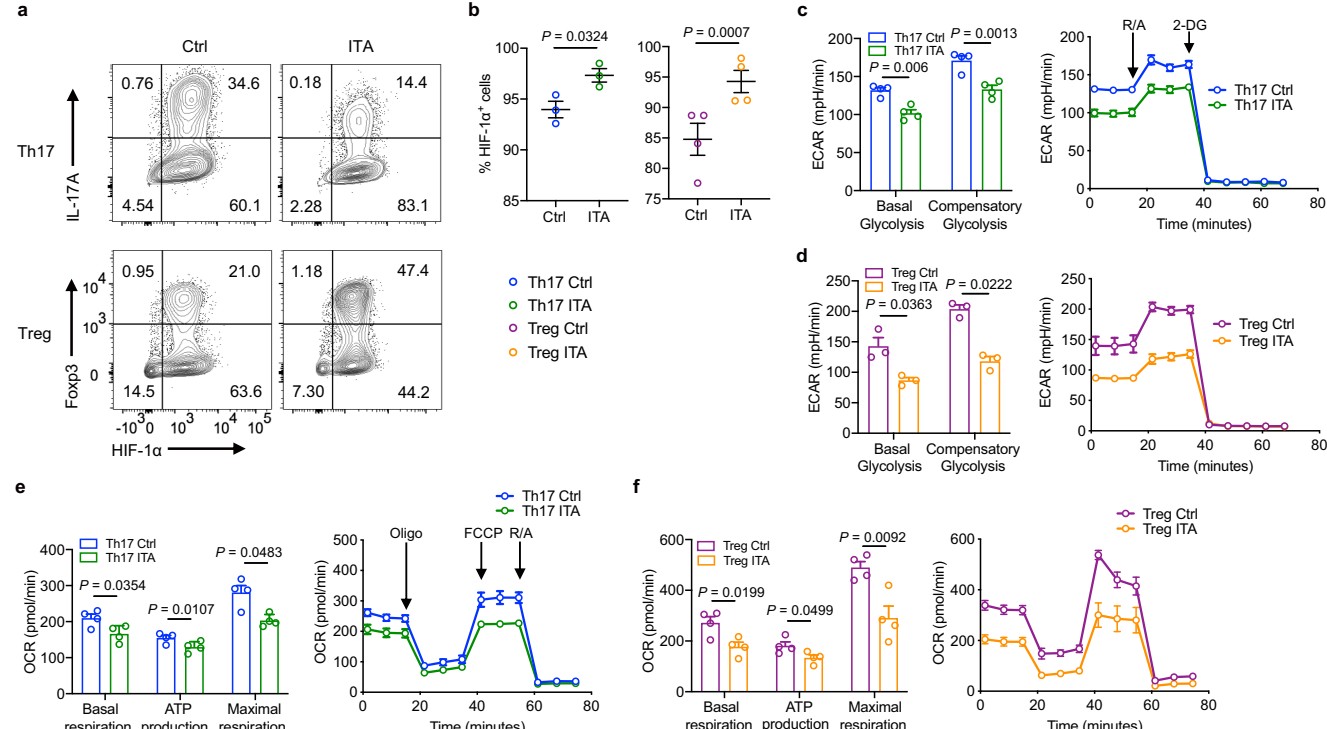

**Fig. 4 | Itaconate inhibits glycolysis without suppressing HIF-1α.** Intracellular HIF-1α expression in Th17- and Treg-polarizing T cells with or without itaconate (ITA) (**a**), percentage of HIF-1α⁺ cells under each growth condition (**b**) (Th17, n = 3; Treg, n = 4). Extracellular acidification rate (ECAR) of Th17- (**c**) and Treg-polarizing T cells (**d**) with or without ITA measured using glycolytic rate assay (Th17, n = 4;

Treg, n = 3). Basal glycolysis and compensatory glycolysis were calculated. Mito-chobdrial oxygen consumption rate (OCR) using an extracellular flux analyzer in Th17- (**e**) and Treg-polarizing T cells (**f**) with or without ITA (n = 4, each condition). *P* values are calculated using two-tailed unpaired Student's *t*-test for (**b**–**f**). Data are representative of mean ± s.e.m. Source data are provided as a Source Data file.

analyzer to assess the metabolic function of T cell differentiation in the presence or absence of ITA. Basal glycolysis and compensatory glycolysis were inhibited in ITA-treated Th17- and Treg-polarizing T cells, as demonstrated by extracellular acidification rate (ECAR) (Fig. 4c, d). Given that ITA inhibits the enzymatic activity of succinate dehydrogenase (SDH) in macrophages[28], we also evaluated mitochondrial OXPHOS. The analysis of mito-chondrial oxygen consumption rate (OCR) showed decreased basal and maximal respiration and ATP production in ITA-treated Th17- and Treg-polarizing T cells (Fig. 4e, f). Overall, these results indicate that ITA inhibits glycolysis and OXPHOS in Th17- and Treg-polarizing T cells.

Unique cytokine signals direct metabolic changes and dis-tinct CD4⁺ T cell subsets differentiation[4]. To investigate whether the effect of ITA on T cell differentiation is dependent on the cytokine signals, we performed RNA-seq using Th0 cells with or without ITA treatment. We identified 1461 DEGs between ITA-treated and control T cells under Th0 condition (Supplementary Fig. 4a). In KEGG pathway analysis, these genes presented main pathways different from those presented by Th17 and Treg RNA-seq data (Supplementary Fig. 4b). The absence of glycolysis and HIF-1 signaling pathway in the overlap between DEGs between ITA-treated and control T cells under Th0, Th17, and Treg con-ditions suggested that ITA strongly induced these metabolic changes dependent on unique cytokine signals for Th17 or Treg differentiation (Supplementary Fig. 4c). Furthermore, pre-treatment of Th0 cells with ITA followed by polarization in Th17 or Treg conditions induced no significant change in Th17 differ-entiation and a slight increase in Treg differentiation (Supple-mentary Fig. 4d, e). These findings indicate that the coordination between ITA treatment and unique cytokine signals is required to regulate Th17/Treg differentiation in ITA-treated T cells.

## Itaconate induces key metabolic changes by inhibiting MAT and IDH1/2 enzymatic activity

To further unravel metabolic profiling, we performed a metabolomic analysis using capillary electrophoresis time-of-flight mass spectro-metry (CE-TOF MS) to evaluate intracellular metabolites in ITA-treated Th17- and Treg-polarizing T cells (Fig. 5a and Supplementary Fig. 5a, b). ITA treatment in vitro increased intracellular levels of ITA in activated naïve CD4⁺ T cells after CD3/CD28 stimulation (Supplementary Fig. 5c). Consistent with the RNA-seq results, the levels of glycolysis pathway metabolites were markedly decreased in ITA-treated Th17- and Treg-polarizing cells (Fig. 5b).

The universal methyl donor SAM, which is synthesized from methionine by an enzyme methionine adenosyltransferase (MAT), regulates gene expression by modulating histone methylation via histone methyltransferases[29]. Decreased SAM levels induced the demethylation of histone H3K4 trimethylation (H3K4me3) in the *Il17a* promoter, resulting in the downregulation of *Il17a* gene expression in Th17 cells[13]. We identified that ITA treatment significantly decreased the level of SAM (Fig. 5a). Interestingly, the methylation index, calcu-lated as the SAM to S-adenosylhomocysteine (SAH) ratio, which is an indicator of cellular methylation potential[13], was reduced in ITA-treated Th17-polarizing T cells, but not in ITA-treated Treg-polarizing T cells (Fig. 5c). Further supporting our results, ITA-treated Th17 cells exhibited a similar trend towards the change of Th17 signature genes (Supplementary Fig. 3c), which were observed in Th17 cells cultured under methionine restriction[13]. The accumulation of 2-HG, synthesized by wild-type isocitrate dehydrogenase (IDH)1 and IDH2, reduces FOXP3 levels in T cells[14]. Similarly, the knockdown of both *IDH1* and *IDH2* reduces the production of 2-HG and increases the expression of FOXP3[14]. We found that ITA reduced the level of 2-HG in Treg-polarizing T cells but not in Th17-polarizing T cells (Fig. 5a). We examined whether increasing intracellular SAM or 2-HG restored the

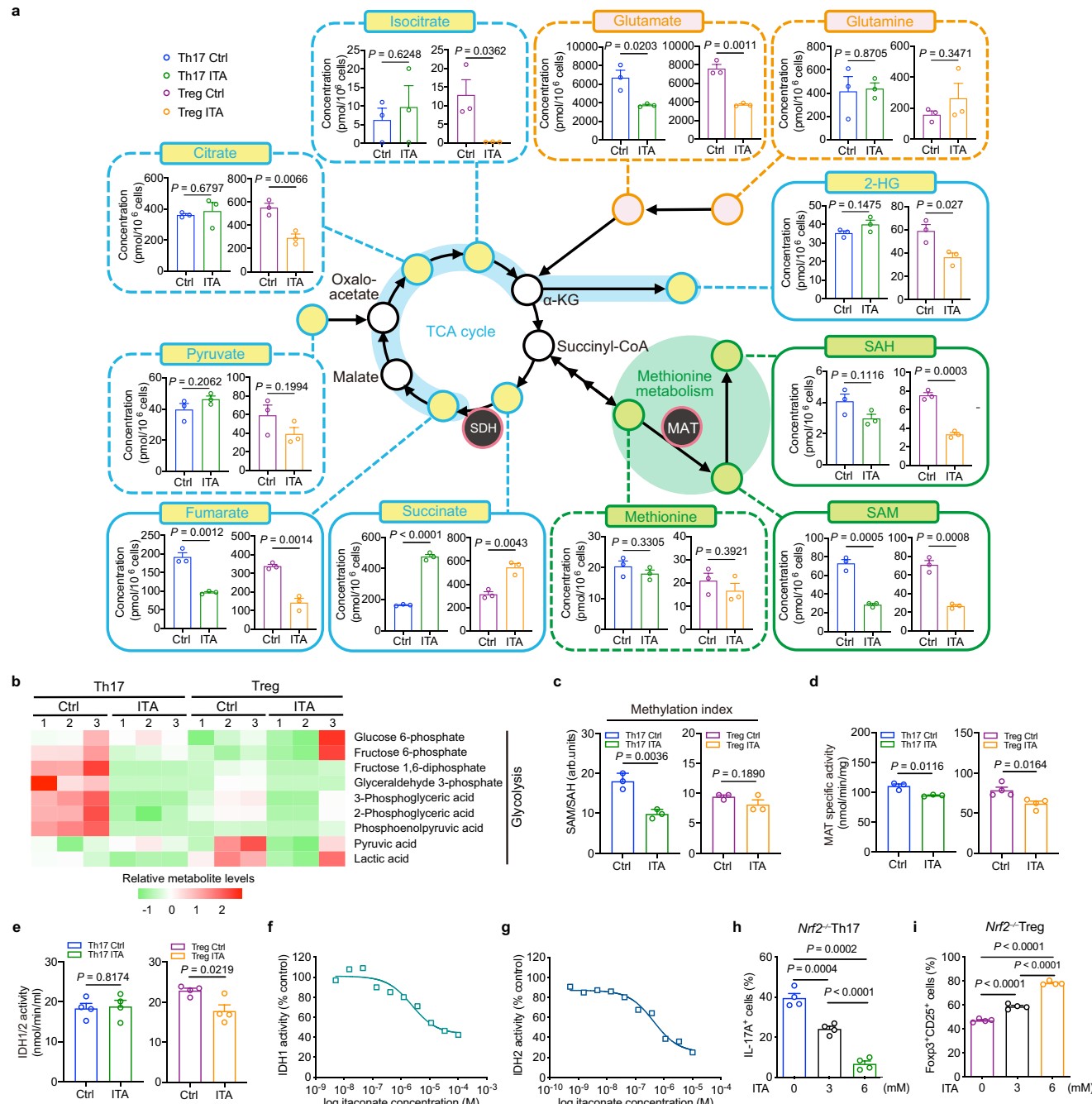

**Fig. 5 | Metabolic reprogramming and enzymatic inhibition in Itaconate-treated Th17- and Treg-polarizing T cells. a** Intracellular levels of metabolites implicated in TCA cycle, glutaminolysis, and methionine metabolism from Th17- and Treg-polarizing T cells in the presence or absence of itaconate (ITA) after 2 days of culture ($n = 3$ independent experiments). **b** Heatmap of glycolytic metabolites from Th17- and Treg-polarizing T cells in the presence or absence of ITA. **c** Methylation index (SAM/SAH ratio) of Th17- and Treg-polarizing T cells in the presence or absence of ITA ($n = 3$, each condition). Enzymatic activity of methionine adenosyltransferase (MAT) (**d**) (Th17, $n = 3$; Treg, $n = 4$) and isocitrate dehydrogenase (IDH)1 and 2 (**e**) in Th17- and Treg-polarizing T cells with or without ITA ($n = 4$, each condition). IDH1 (**f**) and 2 (**g**) activity were measured in the presence of a variable concentration of ITA. IDH activity was calculated as the ratio of measured data to the average control value. Cumulative data of the differentiation of murine naive CD4$^+$ T cells from Nrf2-knockout mice activated under Th17- (**h**), and Treg cell conditions (**i**) in the presence or absence of ITA ($n = 4$, each condition). $P$ values are calculated using two-tailed unpaired Student's $t$-test for (**a, c–e**) and one-way ANOVA with Bonferroni post hoc test for (**h, i**). arb.units, arbitrary unit. Data are representative of mean ± s.e.m. Source data are provided as a Source Data file.

effect of ITA on Th17 or Treg cell differentiation, respectively. Intracellular SAM is derived from extracellular methionine in Th17-polarizing T cells[13]. Treatment with escalating doses of methionine gradually promoted Th17 differentiation in ITA-treated Th17-polarizing T cells (Supplementary Fig. 5g). Additionally, increasing doses of cell-permeable 2-HG impaired Treg differentiation in ITA-treated Treg-polarizing T cells (Supplementary Fig. 5h). These data indicate that the SAM or 2-HG levels affect the polarization programs of ITA-treated Th17 or Treg cells, respectively.

To understand the functional relevance of the altered metabolic intermediates, including SAM and 2-HG, the activities of the synthetic enzymes MAT and IDH1/2 were evaluated. In line with the metabolic

profiling analysis, the enzymatic activity of MAT was inhibited in ITA-treated Th17- and Treg-polarizing cells (Fig. 5d and Supplementary Fig. 5d). Besides, the enzymatic activities of IDH1/2 were inhibited in Treg-polarizing T cells, but not in ITA-treated Th17-polarizing T cells upon ITA treatment (Fig. 5e and Supplementary Fig. 5e). A previous study has shown that ITA binds directly to TET-family DNA dioxygenases like the co-substrate α-ketoglutarate, and inhibits its catalytic activity[30]. Since isocitrate, a co-substrate of IDH, has a similar structure to ITA, we speculated that ITA directly inhibits IDH activity. As shown in Fig. 5f, g, our results supported this speculation and showed that ITA directly inhibited the activity of purified wild-type IDH1 and 2. Further, we prepared whole-cell extract including IDH from Th17- and Treg-polarizing T cells and demonstrated that ITA inhibits the IDH activity of the extract derived from Treg but not Th17 (Supplementary Fig. 5f). Our metabolomics data revealed that ITA treatment led to increased succinate and decreased fumarate levels in T cells, suggesting an inhibitory effect on SDH (Fig. 5a). In the TCA cycle, the metabolite gradients generated by SDH inhibition were preserved and led to decreased isocitrate levels in ITA-treated Treg cells alone after the influx of metabolic components such as pyruvate (Fig. 5a). Based on these data, we inferred that the difference in the co-substrate levels of each cell may affect the inhibitory effect of ITA in IDH enzyme-catalyzed reaction. Altogether we demonstrated the functional link between the metabolic profile and enzymatic activities, indicating that ITA modulates the balance between Th17 and Treg cell differentiation via interaction with key enzymes and metabolites.

ITA exerts anti-inflammatory effects via activation of Nrf2 in macrophages[18]. We quantified the expression of Nrf2 (*Nfe2l2*) and *Hmox1*, a prototypical Nrf2 target gene[31], in Th17- and Treg-polarizing T cells with or without ITA. ITA showed a trend toward increased gene expression of *Nfe2l2*; however, the expression of the downstream target gene transcripts was not significantly increased (Supplementary Fig. 5i). Immunoblotting exhibited that the protein levels of NRF2 and heme oxygenase 1 (HMOX1) did not increase in ITA-treated Th17- and Treg- polarizing T cells compared to those in the control (Supplementary Fig. 5j–l). Nrf2 is also known to facilitate glutaminolysis and redirect glutamate into anabolic pathways[32]. We assessed whether Nrf2 contributes to ITA-mediated regulation of Th17/Treg cell differentiation using Nrf2-deficient mice. We demonstrated that ITA inhibited Th17 cell differentiation and promoted Treg cell differentiation in Nrf2-deficient T cells (Fig. 5h, i). These results suggest that Nrf2 activation is unlikely to be the main mechanism of the regulation of Th17/Treg cell differentiation by ITA.

**Metabolic changes were associated with altered chromatin accessibility of essential transcription factors in Th17 and Treg cell differentiation**

Histone modifications can alter the accessibility of transcription factors to certain genomic regions[33]. Although ITA increased RORγt and HIF-1α expression, a significant decrease in IL-17A expression was observed (Figs. 1b, d and 4a, b). Therefore, we hypothesized that ITA alters chromatin accessibility at key gene loci. To test this hypothesis, we examined RORγt-binding to the *Il17a* promoter in ITA-treated Th17-polarizing T cells using chromatin immunoprecipitation (ChIP) analysis. Naive CD4+ T cells were analyzed as a negative control. Our results showed that ITA suppressed RORγt binding to the *Il17a* promoter (Fig. 6a). During Th17 differentiation, multiple transcription factors are required for the induction of RORγt and IL-17A. In TCR-activated CD4+ T cells, BATF and IRF4 bind cooperatively to the regions of *Il17a* loci. Following Th17-polarizing cytokine activation, HIF-1α, STAT3, and Runx1 are recruited to the same regions. Finally, the RORγt binding to the regions determines *Il17a* gene expression[34]. Because multiple transcription factors are involved in T cell differentiation, we performed an assay for transposase-accessible chromatin sequencing (ATAC-seq) to reveal whether ITA

changes chromatin accessibility of the transcription factors in Th17- and Treg-polarizing T cells. As RNA expression depends on DNA accessibility, we integrated the ATAC-seq and RNA-seq datasets (Fig. 6b). Overall, 703 DEGs showed differential accessibility between ITA-treated and control T cells under Th17 or Treg conditions. These genes included *Il17a*, glycolysis-related genes, and *foxp3*, but not *csf2*, which encodes GM-CSF (Fig. 6c, d). We next focused on two groups: chromatin more closed in ITA-treated Th17 cells, and chromatin more open in ITA-treated Treg cells. The analysis of transcription factor binding motifs revealed that the peaks of Th17 group were enriched for the motifs regulating *Il17a* gene expression, including BATF, STAT3, IRF4, RUNX1, and HIF-1α (Fig. 6e). In addition, motif discovery also indicated that the peaks of Treg group were enriched for ETS1, RUNX1, SMAD3, STAT5, and AP-1 (Fig. 6f), suggesting that ITA mainly altered the accessibility of conserved non-cording sequence 2 (CNS2)-targeting transcription factors which maintain *Foxp3* expression[35]. Collectively, these results demonstrated that ITA treatment leads the chromatin accessibility to closed in *Il17a* loci and open in *Foxp3* loci for key transcription factors.

## Discussion

Our study revealed that ITA is a key metabolite in metabolic and epigenetic reprogramming to suppress Th17 and promote Treg cell differentiation. Consistently, the adoptive transfer of ITA-treated Th17-polarizing T cells ameliorated EAE.

Activated CD4+ T cells require metabolic and epigenetic reprogramming for proliferation and differentiation from the naive state[4]. Several studies have shown that multiple metabolic pathways, including glycolysis, glutaminolysis, and one-carbon metabolism, induce the reprogramming and control of Th17/Treg cell differentiation[13,14,36]. However, it remains unclear whether these processes are regulated independently of one another. Although the mammalian target of rapamycin (mTOR)-HIF-1α pathway is a crucial process for integrating glycolysis and Th17/Treg cell differentiation[27], our results showed that ITA inhibits glycolysis without suppressing HIF-1α. Following inhibition of discrete metabolism, the intracellular metabolites, such as SAM and 2-HG, influence Th17 or Treg cell differentiation by epigenetic reprogramming[13,14]. Mechanistically, the reduced SAM/SAH ratio in Th17 cells induced histone demethylation not only H3K4me3 but also histone marks for transcriptional repression, including H3K27me3[13]. Decreased 2-HG induced hypomethylation at the *Foxp3* promoter and CNS2[14]. Given that individual histone marks have discrete regulatory roles[33], it is complicated how histone modification by altering the metabolites contributes to gene expression. Our study revealed that ITA decreased SAM/SAH ratio and 2-HG by inhibiting MAT and IDH1/2, respectively. Consequently, ITA altered the chromatin accessibility of essential transcription factors at the *Il17a* and *Foxp3* loci, resulting in suppressed IL-17A and increased FOXP3 expression.

In this study, we showed the modulatory role of ITA in mediating the balance of Th17/Treg cell differentiation. The high polarity and low electrophilicity of ITA result in its low cell permeability[37]. To further confirm the mechanism of ITA uptake in T cells, a transport-mediated process such as mitochondrial oxoglutarate, dicarboxylate, and citrate carriers[18] should be assessed. We focused on the discrepancy that RORγt expression was increased with ITA, but IL-17A expression was decreased. Concordant with our study, a previous study has reported that Th17 cells cultured under methionine restriction, which induces SAM reduction, showed reduced IL-17 production but stable RORγt expression[13]. Decreased SAM levels induced demethylation of histone H3K4me3 at the *Il17a* promoter, resulting in the downregulation of *Il17a* gene expression in Th17 cells. In the same study, SAM reduction also resulted in the demethylation of H3Kme3 around the *Rorc* transcription start site, but this did not decrease *Rorc* gene expression. Thus, the influence of SAM reduction

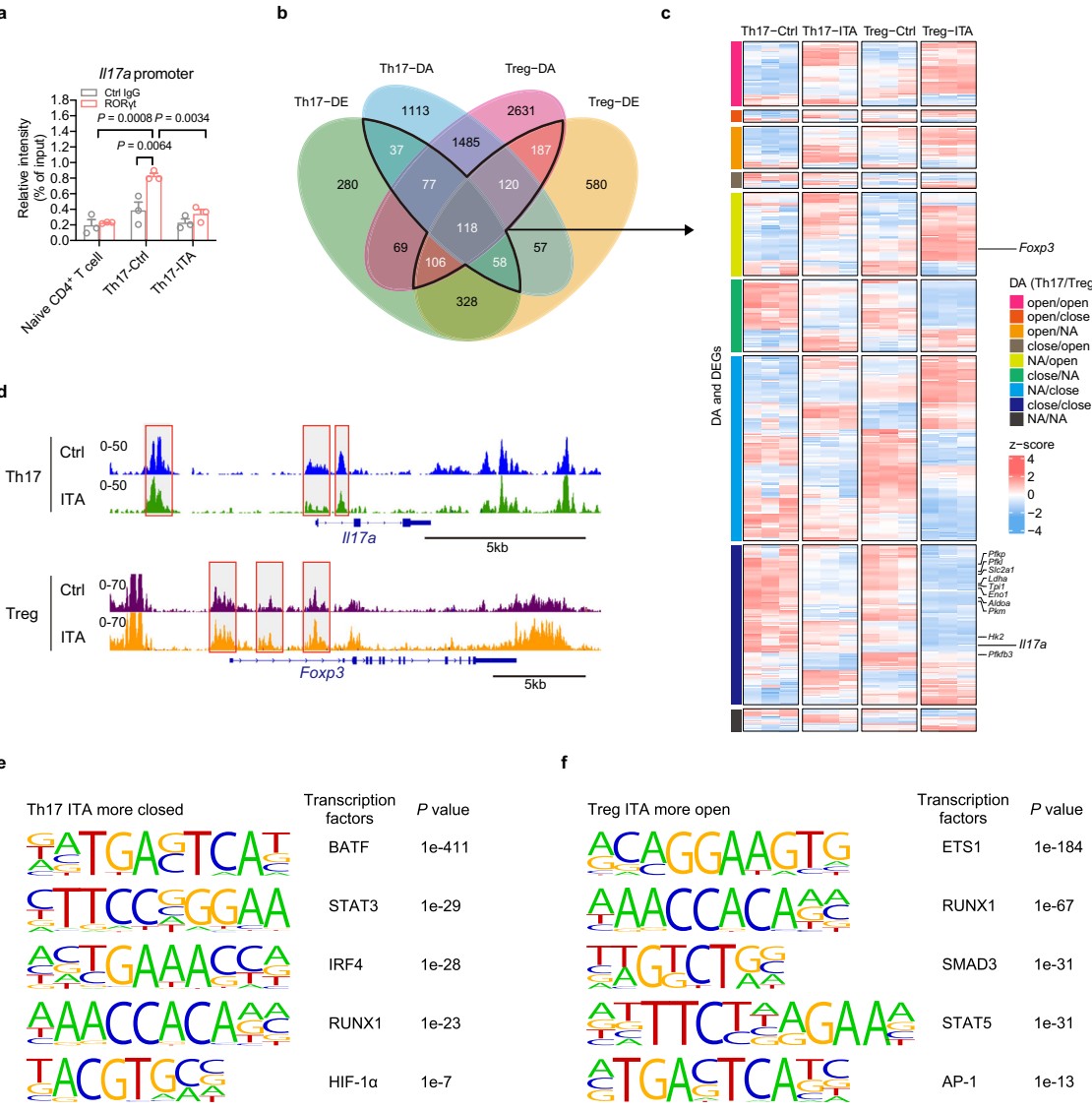

**Fig. 6 | Itaconate altered the chromatin accessibility of essential transcription factors in Th17 and Treg cell differentiation. a** Chromatin immunoprecipitation (ChIP) analysis of RORγt at the *Il17a* promoter regions in naive CD4+ and Th17-polarizing T cells in the presence or absence of itaconate (ITA) (Naive CD4+, *n* = 3; Th17, *n* = 3). **b**–**f** Assay for transposase-accessible chromatin sequencing (ATAC-seq) was performed using Th17- and Treg-polarizing T cells from B6 mice in the presence or absence of ITA after 2 days of culture. Two replicates (*n* = 2, each group) were used for ATAC-seq. Venn diagram displaying the overlap between DEGs between ITA-treated and control T cells under Th17 or Treg conditions and between genes that show differential chromatin accessibility (DA) (**b**). Heatmap displaying relative expression (z-score) of genes that show DA and DE between ITA-treated and control T cells under Th17 or Treg conditions (**c**). **d** Representative ATAC-seq tracks in Th17- and Treg-polarizing T cells. **e** Motif discovery of the peaks which significantly changed to be more closed in ITA-treated Th17 cells compared to Ctrl. **f** Motif discovery of the peaks which significantly changed to be more open in ITA-treated Treg cells compared to Ctrl. *P* values are calculated using one-way ANOVA with Bonferroni post hoc test for (**a**) and two-sided Wald test with Benjamini and Hochberg method for (**e**, **f**). Data are representative of mean ± s.e.m. Source data are provided as a Source Data file.

on the histone modification may not be specific to *Il17a* gene loci, and additional mechanisms may affect *Rorc* expression. Further studies are required to reveal this selectivity of gene expression. In the motif analysis, RORγt was not detected in the Th17 group. Reportedly, RORγt possesses some binding sites other than at the promoter region of *Il17a*[38], and the promoter region is marked by H3K4me3[39], the most labile mark affected by the restriction of the methionine cycle[13]. Therefore, we speculated that ITA might not change the accessibility of RORγt at the *Il17a* locus other than at the promoter region, and the accessibility in these RORγt-binding sites does not appear to change. Although our work focused on key metabolites and the binding of essential transcription factors at the *Il17a* and *Foxp3* loci, the findings indicate that ITA may regulate other metabolites or transcription factors that impact T cell differentiation.

As multiple pathways are involved in the anti-inflammatory effect of ITA on macrophages[18,19], unknown mechanisms may contribute to the regulation of ITA-mediated T cell differentiation. In addition, future studies should address the reliable delivery of ITA to T cells for translation of its use in a clinical context.

In summary, we identified ITA as a key regulator reducing Th17 cell differentiation and promoting Treg cell differentiation through metabolic and epigenetic reprogramming. Our results could integrate previous knowledge of key metabolites and epigenetics in T cells and offer mechanisms and options for the modulation of T cell differentiation. Given the pathogenic roles of Th17/Treg imbalance in a wide variety of autoimmune diseases, our study makes a worthwhile contribution to suggesting simple therapeutic approaches which regulate T cell differentiation.

## Methods

### Mice

C57BL/6J mice were purchased from Charles River Laboratories (Wilmington, MA). 2D2 (C57BL/6-Tg(Tcra2D2, Tcrb2D2)1Kuch/J) and $Rag1^{-/-}$ (B6.129S7-Rag1$^{tm1Mom}$/J) were purchased from Jackson Laboratories (Bar Harbor, ME). Nrf2$^{-/-}$ (B6.129P2-Nfe2l2$^{tm1Mym}$/MymRbrc) mice were provided by the RIKEN BRC through the National BioResource Project of the MEXT/AMED, Japan. All mice were bred in house and maintained in temperature- and humidity-controlled facilities under pathogen-free conditions at the Hokkaido University, group housed with free access to food and water and 12 h light/dark cycles. Both male and female mice were used at 8–10 weeks old with age- and sex-matched controls. All animal experiments were approved by the Institutional Animal Care and Use Committee of Hokkaido University (permission number: 19-0147).

### T cell in vitro activation and culture

Naive CD4$^+$ T cells were isolated from the murine spleen by magnetic cell sorting with the naive CD4$^+$ T Cell Isolation kit (Miltenyi Biotec). Approximately 0.3 million naive CD4$^+$ T cells were plated into 48-well-plate pre-coated with goat anti-hamster IgG (MP Biomedicals) and stimulated for 2–3 days with anti-CD3 (0.25 µg mL$^{-1}$, clone 145-2C11, Biolegend, 100340) and anti-CD28 (0.5 µg mL$^{-1}$, clone 37.51, Biolegend, 102116) antibodies[40]. For each T cell differentiation, subset-specific antibodies and cytokines were further supplemented. For Th1 differentiation, cells were cultured with anti-IL-4 antibody (3 µg mL$^{-1}$, clone 11B11, Biolegend, 504122) and IL-12 (20 ng mL$^{-1}$, Biolegend, 577004). For Th2 differentiation, cells were cultured with anti-IFNγ antibody (3 µg mL$^{-1}$, clone AN-18, Biolegend, 517906) and IL-4 (100 ng mL$^{-1}$, Biolegend, 574304). For Th17 differentiation, cells were cultured with anti-IL-4 antibody (2 µg mL$^{-1}$, clone 11B11, Biolegend, 504122), anti-IFNγ antibody (2 µg mL$^{-1}$, clone AN-18, Biolegend, 517906), IL-6 (30 ng mL$^{-1}$, Biolegend, 575704), and TGF-β (0.3 ng mL$^{-1}$, Miltenyi Biotec, 130-095-066). For Treg differentiation, cells were cultured with anti-IL-4 antibody (2 µg mL$^{-1}$, clone 11B11, Biolegend, 504122), anti-IFNγ antibody (2 µg mL$^{-1}$, clone AN-18, Biolegend, 517906), IL-2 (20 ng mL$^{-1}$, Biolegend, 575404), and TGF-β (1 ng mL$^{-1}$, Miltenyi Biotec, 130-095-066). All cells were cultured in RPMI 1640 medium containing 10% FBS, 0.1% 2-mercaptoethanol, and penicillin-streptomycin at 37 °C under 5% $CO_2$. Itaconate (ITA, Sigma-Aldrich, I29204), L-Methionine (Sigma-Aldrich, M5308), and disodium (R)−2-hydroxtglutarate (2-HG, Selleck, S7873) were prepared as 750, 250, 100 mM stock solutions in PBS, respectively, and diluted directly into culture media at various concentrations. The medium, including ITA, L-Methionine, and 2-HG, was adjusted to pH 7.4 with 1 N NaOH at 37 °C.

### BMDMs culture

Bone marrow cells were harvested from the femur and tibia of C57BL/6 J mice and differentiated in the presence of M-CSF (20 ng mL$^{-1}$, R&D Systems, 416-ML-010) in RPMI 1640 medium containing 10% FBS, 0.1% 2-mercaptoethanol, and penicillin-streptomycin at 37 °C under 5% $CO_2$ for 8 days. On day 8, the bone-marrow-derived macrophage (BMDMs) were washed and stimulated with lipopolysaccharide (LPS, 100 ng mL$^{-1}$, Sigma-Aldrich, L2880) for 6 h.

### Immunoblotting

Cultured T cells were dissolved in lysis buffer (FUJIFILM Wako, 038-21141) supplemented with protease inhibitor (Sigma-Aldrich, P8340). The lysates were boiled at 95 °C for 5 min in Laemmli sample buffer. They were then resolved on 4–15% SDS-PAGE gel electrophoresis and transferred to a polyvinylidene difluoride membrane. The membranes were then blocked for 30 min with 2.5% nonfat milk in TBS-Tween for 30 min at 37 °C. After blocking, the membranes were incubated overnight at 4 °C with the following primary antibodies: anti-mouse NRF2 (clone D1Z9C, Cell Signaling Technology, 12721S; 1:500), anti-mouse

HO-1 (HMOX1, clone E6Z5G, Cell Signaling Technology, 82206; 1:500), and anti-β-actin (clone AC-15, Sigma-Aldrich, A3854; 1:50,000); β-actin was used as an endogenous control. Afterward, horseradish peroxidase-conjugated anti-rabbit (Sigma-Aldrich, A0545; 1:80,000) and anti-mouse (Sigma-Aldrich, A9044; 1:120,000) antibodies were used as secondary reagents. Signals were detected using enhanced chemiluminescence western blot detection reagents (Cytiva, RPN2209) and the LAS-4000 imaging system (FUJIFILM, Japan).

### Flow cytometry

Cells were collected from in vitro culture after 3 days post-activation or in vivo mice experiments. For intracellular staining, cells were re-stimulated with phorbol 12-myristate 13-acetate (PMA: 500 ng mL$^{-1}$, Sigma-Aldrich, P1585) and ionomycin (1 µg mL$^{-1}$, Sigma-Aldrich, I19657) under GolgiStop (BD Biosciences, 554724), except for Foxp3 detection. After 4 h, cells were stained with Zombie Aqua Fixable Viability Kit (Biolegend, 423102) to eliminate dead cells and anti-CD16/CD32 mix (0.01 mg mL$^{-1}$, clone 2.4G2, BD Biosciences, 553141) to prevent nonspecific binding. Cells were fixed and permeabilized with fixation and permeabilization solution (BD Biosciences, 554715), followed by staining with cytokine- and/or transcription factor-specific antibodies as follows: anti-mouse CD4 (clone H129.19, Biolegend, 130308; 1:100), CD3 (clone 17A2, Biolegend, 100217; 1:100), CD90.2 (clone 53-2.1, Biolegend, 140310; 1:100), CD11b (clone M1/70, BD Biosciences, 557657; 1:100), Ly6C (clone HK1.4, Biolegend, 128006; 1:100), Ly6G (clone 1A8, Biolegend, 127613; 1:100), B220/CD45R (clone RA3-6B2, Biolegend, 563103; 1:100), F4/80 (clone BM8, Biolegend, 123113; 1:100) antibodies, for surface, and anti-mouse IFN-γ (clone XMG1.2, Biolegend, 505830; 1:50), IL-4 (clone 11B11, Biolegend, 504104; 1:50), IL-17A (clone TC11-18H10.1, Biolegend, 506904; 1:50), RORγt (clone B2D, eBioscience, 17-6981-80; 1:50), GM-CSF (clone MP1-22E9, eBioscience, 17-7331-82; 1:50), HIF-1α (clone 241812, R&D Systems, IC1935A; 1:50), and pro-IL-1β (clone NJTEN3, eBioscience, 12-7114-80; 1:50) antibodies, for intracellular. For Treg staining, cells were stained with anti-mouse CD25 (for surface, clone 3C7, BD Biosciences, 564370; 1:100), Foxp3 (for intracellular, clone FJK-16s, eBioscience, 12-5773-82; 1:50), and HIF-1α (for intracellular, clone 241812, R&D Systems, IC1935A) antibodies. All flow cytometry data were acquired on a BD FACSAria III cell sorter (BD Biosciences) and analyzed using the Flowjo software.

### Cell proliferation assay

Naive CD4$^+$ T cells were labeled with CSFE (1 µM, BD Biosciences, 565082) and then cultured under Th17 or Treg cell conditions for 3 days before FACS analysis.

### Active EAE model

EAE was induced by immunizing C57BL/6J mice with myelin oligodendrocyte glycoprotein (MOG)$_{35-55}$ peptide (100 µg per mouse, ANASPEC, 60130-5) and complete Freund's adjuvant (CFA) containing 4 mg ml$^{-1}$ (0.4 mg per mouse) of heat-killed *Mycobacterium tuberculosis* (Chondrex, 7001)[24]. Mice were intraperitoneally injected with pertussis toxin (PTX, 250 ng/mouse, List Biological Laboratories, 180) on days 0 and 2. PBS or ITA (50 mg kg$^{-1}$) was intraperitoneally injected every other day from day 0 to day 14. An ITA solution was adjusted to pH 7.4 with 1 N NaOH at 37 °C. The disease scores were assigned according to the following scale: 0, no clinical signs; 0.5, partially limp tail;1, paralyzed tail; 2, loss in coordinated movement, hind limb paresis; 2.5; one hind limb paralyzed; 3, both hind limbs paralyzed; 3.5, hind limbs paralyzed, weakness in forelimbs; 4, forelimbs paralyzed; 5, moribund or death[40].

### Passive transfer EAE model

On day 0, naive CD4$^+$ T cells were isolated from 8–10-week-old 2D2 mice, as described above. Approximately 0.2 million naive CD4$^+$ T cells

were plated into 48-well-plate pre-coated with goat anti-hamster IgG (MP Biomedicals) and stimulated for 3 days with anti-CD3 (0.25 μg mL⁻¹, clone 145-2C11, Biolegend) and anti-CD28 (0.5 μg mL⁻¹, clone 37.51, Biolegend) antibodies[12]. Cells were cultured under Th17-polarizing conditions with anti-IL-4 (2 μg mL⁻¹, clone 11B11, Biolegend), anti-IFNγ (2 μg mL⁻¹, clone AN-18, Biolegend), IL-6 (30 ng mL⁻¹, Biolegend), IL-23 (10 ng mL⁻¹, Biolegend, 589004), IL-1β (10 ng mL⁻¹, Biolegend, 575104), and TGF-β (0.3 ng mL⁻¹, Miltenyi Biotec) with or without ITA (3 mM, Sigma-Aldrich). Cultured cells were purified on day 3 of culture, and $1 \times 10^7$ cells were injected intravenously into each *Rag1*-deficient recipient mouse (8–10-week-old). Pertussis toxin (400 ng per mouse, List Biological Laboratories, 180) was intraperitoneally injected on the day of transfer and two days later. After 14 days of EAE induction, the mice were anesthetized, and the spinal cords were collected for analysis.

For histological staining, sections from 10% formalin-fixed spinal cords were stained using H&E and luxol fast blue. The specimens were examined using the Keyence BZ-X700/710 microscopy. Images were analyzed with the Keyence BZ-X Analyzer software. Histology was scored by an investigator blinded to the experimental group. Spinal cord sections were scored as follows: 0, no infiltration (<50 cells); 1, mild infiltration (50–100 cells); 2, moderate infiltration (100–150 cells); 3, severe infiltration (150–200 cells); and 4, massive infiltration (>200 cells)[40].

## RNA isolation and quantitative PCR

Total RNA was extracted from BMDMs on day 8 and from Th0, Th1, Th2, Th17, and Treg cells on day 2 using RNeasy Plus Micro Kit (QIAGEN) according to the manufacturer's instructions. For reverse transcription quantitative PCR (qPCR), isolated RNA was reverse transcribed, and gene expression was quantified using predesigned TaqMan Gene Expression Assays. The primers used were as follows: TaqMan probe *Foxp3* (Mm00475162_m1), TaqMan probe *Rorc* (Mm01261022_m1), TaqMan probe *Il17a* (Mm00446973_m1), TaqMan probe *Il17f* (Mm00521423_m1), TaqMan probe *Irg1* (Mm01224532_m1), TaqMan probe *Nfe2l2* (Mm00477784_m1), TaqMan probe *Hmox1* (Mm00516005_m1), TaqMan probe *Tbp* (Mm00446973_m1), and TaqMan probe *Gusb* (Mm1197698_m1). Gene expression was normalized to the reference gene *Tbp* and *Gusb*.

## RNA-seq analysis

Total RNA freshly isolated from Th0, Th17, and Treg cells with or without ITA treatment on day 2 was extracted using the RNeasy Plus Micro Kit, and subjected to RNA-seq analysis. Total RNA was quantified and qualified using NanoDrop and Qubit RNA Assay (Thermo Fisher Scientific) and RNA ScreenTape on TapeStation 4200 (Agilent Technologies, Palo Alto, CA, USA). Approximately 500 ng total RNA with RNA integrity number 7 or higher was used for poly-A mRNA enrichment (NEBNext Poly(A) mRNA Magnetic Isolation Module). The library was prepared using the NEBNext Ultra II Directional RNA Library Prep Kit for Illumina (New England Biolabs), according to the manufacturer's protocol. The resultant libraries were then quantified and sequenced by Illumina NovaSeq according to the manufacturer's instructions (Illumina, San Diego, CA, USA) with a 150 bp paired-end configuration. The library preparation and sequencing were performed by GENEWIZ. The raw sequencing reads with low quality and adapter sequences were removed using Cutadapt v 2.1. The trimmed reads were mapped to the mm10 (Ensembl 99) and quantified using STAR v 2.7.1a[41]. Data analysis was performed using R platform v 3.6.1. DEGs were identified based on differences in expression levels (log2 fold-change > 0.5 and adjusted *p* value < 0.01) between samples after removing genes with zero read count using DESeq2 v 1.24.0[42]. The Benjamini-Hochberg method was used to adjust the *p* value for multiple hypothesis testing. Gene ontology (GO) and Kyoto Encyclopedia

of Genes and Genomes (KEGG) pathway analysis of DEGs was performed using Metascape v 3.5 web-based platform[43].

## Extracellular flux analysis

The Seahorse XFp Extracellular Flux Analyzer (Agilent Technologies) was used to measure ECAR and OCR[36]. Briefly, naive CD4⁺ T cells isolated from WT mice were cultured under Th17- and Treg-polarizing conditions, with or without ITA. After 2 days, $1.5 \times 10^5$ cells were plated in an XFp microplate coated with cell-tak (Corning). The Glycolytic Rate Assay (Agilent Technologies) was performed to measure ECAR, followed by sequential injections of rotenone/antimycin A (Rot/AA: 0.5 μM) and 2-deoxyglucose (50 mM). The Cell Mito Stress test (Agilent Technologies) was performed to measure OCR, followed by sequential injections of Oligomycin (1.5 μM), FCCP (0.5 μM), and Rot/AA (0.5 μM). The obtained data were analyzed by Seahorse Wave v 2.6.0 (Agilent technologies).

## Profiling of intracellular metabolites

Intracellular metabolites were extracted from $14 \times 10^6$ cells for each sample. Cells were collected by centrifugation ($800 \times g$ and 4 °C for 2 min) and washed with 10 mL 5% mannitol solution, then treated with 800 μL methanol and vortexed for 30 s to inactivate enzymes. Next, the cell extract was treated with 550 μL Milli-Q water containing internal standards (Human Metabolome Technologies) and left to rest for another 30 s. The extract was obtained and centrifuged at $2300 \times g$ and 4 °C for 5 min. Then, 800 μL upper aqueous layer was centrifugally filtered through a Milli-pore 5-kDa cutoff filter at $9100 \times g$ and 4 °C for 120 min to remove proteins. The filtrate was centrifugally concentrated and resuspended in 50 μL Milli-Q water for CE-TOF MS analysis[44]. Samples were measured and analyzed by Human Metabolome Technologies.

## Enzyme activity assay

Enzymes, including MAT and IDH1/2, were purified from Th17- and Treg-polarizing T cells with or without ITA on day 2 and the activities were assessed by an enzyme activity assay kit (all from Biovision) according to the manufacturer's instructions.

To evaluate the effect of ITA on IDH1 and 2 activities, ten-dose IDH profiling was performed in Reaction Biology Corporation. IDH1 or 2 wild-type enzymes and NADP⁺ (for IDH1; 50 μM NADP⁺, for IDH2; 25 μM NADP⁺) in the reaction buffer (50 mM KH₂PO₄, pH 7.5, 10 mM MgCl₂, 2.5% glycerol, 150 mM NaCl, 0.05% BSA, 2 mM β-ME, 0.003% Brij35) were plated into wells of the reaction plate. ITA in PBS (pH was adjusted to 7.4 with 5 N NaOH) or control compounds were delivered into the enzyme mixture and pre-incubated for 60 min at room temperature. To initiate the enzyme-catalyzed reaction, a substrate mixture (for IDH1, 65 μM Isocitrate; for IDH2, 150 μM Isocitrate) was added and incubated for 60 min at room temperature. A detection mixture was added before measuring the IDH activity by EnVision (Ex/Em = 535/590 nm). An ITA 100 mM stock solution was prepared in PBS and adjusted to pH 7.4 with 5 N NaOH at room temperature.

## ATAC-seq analysis

Cells were harvested and frozen in culture media containing FBS and 5% DMSO. Cryopreserved cells were sent to Active Motif to perform the ATAC-seq assay. The cells were then thawed in a 37 °C water bath, pelleted, washed with cold PBS, and tagmented as previously described[45], with some modifications following a previous study[46]. Briefly, cell pellets were resuspended in lysis buffer, pelleted, and tagmented using the enzyme and buffer provided in the Nextera Library Prep Kit (Illumina). Tagmented DNA was then purified using the MinElute PCR purification kit (Qiagen), amplified with 10 cycles of PCR, and purified using Agencourt AMPure SPRI beads (Beckman Coulter). The resulting material was quantified using the KAPA Library

Quantification Kit for Illumina platforms (KAPA Biosystems) and sequenced with PE42 sequencing on the NextSeq 500 sequencer (Illumina).

Reads were aligned using the BWA algorithm (mem mode; default settings). Duplicate reads were removed, the reads mapping as matched pairs and the uniquely mapped reads (mapping quality ≥ 1) were used for further analysis. Alignments were extended in silico at their 3′-ends to a length of 200 bp and assigned to 32-nt bins along the genome. The resulting histograms (genomic "signal maps") were stored in bigWig files. Peaks were identified using the MACS 2.1.0 algorithm at a cutoff of $p$ value 1e-7, without a control file, and with the '−nomodel' option. Peaks that were on the ENCODE blacklist of known false peaks were removed. Signal maps and peak locations were used as input data to Active Motifs proprietary analysis program, which creates Excel tables containing detailed information on sample comparison, peak metrics, peak locations, and gene annotations. Chromatin accessibility was accessed based on the output table. Differently accessible (DA) peaks were selected with the peak metrics of |shrunkenLog2FC| > 0.3 and adjusted $p$ value < 0.1. DA genes were defined as the genes with DA peaks. The DA genes were classified into "open" or "close" when all of the DA peaks on the annotated genes were in the same direction (i.e., All DA peaks increased on the gene "open").

### ChIP analysis

Freshly isolated naive CD4$^+$ T cells from wild-type mice were cultured under Th17- polarizing conditions, with or without ITA treatment. After 2 days, $4.5 \times 10^6$ cells were harvested, and ChIP was performed using the ChIP system (Thermo Fisher Scientific) according to the manufacturer's instructions. The soluble chromatin supernatant was immunoprecipitated with an anti-RORγt antibody (40 μg mL$^{-1}$, clone AFKJS-9, eBioscience, 14-6988-82) and IgG control (40 μg mL$^{-1}$, clone eBR2a, eBioscience, 14-4321-82). Immunoprecipitated DNA and input DNA were measured by qPCR using a SYBR green reagent. Data were expressed as the percent input for each ChIP fraction. The primers used for amplification of the promoter of *Il17a* containing the RORγt-binding site were 5′ CAGCTCCCAAGAAGTCATGC 3′ (forward) and 5′ GCAACATCTGTCTCGAAGGTAG 3′ (reverse)[47].

### Statistics and reproducibility

Except for RNA-seq and ATAC-seq experiments, statistical analyses were performed with Prism software version 8.0 (GraphPad Prism software) using a two-tailed unpaired Student's $t$-test for pairwise comparison of variables and one-way analysis of variance (ANOVA) with Bonferroni post hoc test for multiple comparisons of variables. For the EAE model, clinical scores and body weight changes of each group were compared using two-way ANOVA. A $P$ value of <0.05 was considered statistically significant. Error bars present the mean ± standard error of the mean (s.e.m.). No statistical methods were used to predetermine sample sizes. Sample sizes were based on pilot experiments conducted in the same laboratory and comparable to similar studies in the field. No data were excluded from the analyses. The evaluation and scoring of histopathology of HE-stained tissue sections was performed in a blinded fashion. In the other experiments, no blinding was used during allocation of experimental groups, because all data collection and analysis is quantitative and not qualitative in nature. To avoid introducing bias, samples were measured in a standardized way.

### Reporting summary

Further information on research design is available in the Nature Portfolio Reporting Summary linked to this article.

## Data availability

RNA-seq data reported in this study are available at the NCBI Gene Expression Omnibus (GEO) under the accession number GSE182895.

ATAC-seq data reported in this study are available at the NCBI GEO under the accession number GSE207941. KEEG pathway analysis in the manuscript indicates the integrated analysis in Metascape. Please find the references to the parent algorithms (Metascape). Source data are provided with this paper.

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

## Acknowledgements

This study was supported by the Japan Agency for Medical Research and Development (AMED) under grant number JP20ek0410078h (to M.Kono), Grants-in-Aid for Regional R&D Proposal-Based Program from Northern Advancement Center for Science & Technology of Hokkaido Japan (to M.Kono), the Kurata Grants from the Hitachi Global Foundation (to M.Kono), Tokyo Biochemical Research Foundation (to M.Kono), and Inamori Foundation (to M. Kono). We thank Ms. Shikishi Chida (Hokkaido University) for her secretary support. We thank BioRender.com for the creation of Supplementary Fig. 2a.

## Author contributions

K.A., M.Kono, and T.A. designed the study. K.A., M.Kono, M.Kanda, Y.K., K.S., and R.H. performed the experiments. K.A., M.Kono, M.Kanda, Y.K., K.S., R.H., K.K., Y.U., D.N., Y.F., M.Kato and T.A. analyzed and interpreted the data. K.A., M.Kono, M.Kanda, Y.K., K.S., R.H., K.K., Y.U., D.N., Y.F., M.Kato, O.A., and T.A. drafted the manuscript. All authors reviewed and approved the manuscript.

## Competing interests

The authors declare no competing interests.
