## [Peer Review File · Nature Communications]

Itaconate ameliorates autoimmunity by modulating T cell imbalance via metabolic and epigenetic reprogrammingREVIEWER COMMENTS

Reviewer #1 (Remarks to the Author):

In previous studies, itaconate, a TCA cycle intermediate, was shown to have anti-inflammatory effects, acting through activation of Nrf2 and, potentially, by inhibiting succinate dehydrogenase, to attenuate cytokine production in macrophages. In this study, the authors show that exogenously administered itaconate inhibits “pathogenic” Th17 differentiation and promotes Tregs. They demonstrate that in vitro exposure to itaconate renders myelin-specific Th17-polarized T cells less pathogenic in the EAE model. They provide evidence that itaconate inhibits glycolysis and also alters the amount of SAM, that, in turn, regulates epigenetic modifications. A potentially interesting finding was the reduced methylation index (SAM/SAH ratio) in itaconate-treated Th17, but not Treg cells, while 2-HG levels were reduced in Treg, but not Th17 cells. The authors link these observations to reduced accessibility of the Il17a locus for transcription factor binding, presumably through less histone methylation, after itaconate treatment.

The mechanism for the anti-inflammatory effect of itaconate has focused on macrophages, and a thorough analysis of its influence on other cells involved in inflammation, including T cells, is needed. Although this paper provides strong data as to the effect of exogenous itaconate on Th17 cell function in vitro along with a plausible explanation for how itaconate may influence the Th17:Treg balance, the relevance in vivo is not addressed, and the quality of some of the data makes it difficult to determine the likelihood of the proposed mechanism. Major points are listed below:

1. The authors describe itaconate as an endogenous metabolite, but test its effects when added exogenously. What is the context in which T cells are exposed to exogenous itaconate rather than having their internal itaconate level modulated?
2. Some of the metabolic effects of itaconate revealed by the authors (such as inhibition of IDH1/2) were described previously in macrophages, and itaconate is known to be a regulator of the TCA cycle.
3. The effects of exogenous itaconate on glycolysis are well demonstrated. However, the analysis of the T cell epigenetic program is superficial, and does not provide much new insight. For example, the ChIP of RORgt shows very little specificity, only 2-fold over IgG, and is difficult to evaluate without other loci being included (both positive and negative loci for binding). The best negative control would be to use RORgt-deficient T cells. It is also difficult to judge from Fig 6b whether there are significant differences in accessibility, as only tracings of the Il17a and Foxp3 loci are shown and there are no statistical analyses based on whole genome profiles of the ATAC-seq. It is important to show other loci that have differences with itaconate, as well as characterize regions showing appearance or disappearance of new/distinct peaks. Is the GM-CSF (Csf2) locus altered? Are there chromatin accessibility changes in glycolysis genes?
4. Performing RNA-seq and/or ATAC-seq and/or metabolomics with itaconate-untreated and -treated naïve T cells under Th0 conditions could provide insight into how the metabolite affects the cellular program in the absence of cytokine signals. Pre-treatment of cells with itaconate followed by polarization in Th17 or Treg conditions could also provide insight into how itaconate functions.

Minor points:

1. How does treatment with itaconate compare to that with dimethyl itaconate, which has been used by other groups because it is more membrane-permeable?
2. While the authors show that cell viability is not affected in Fig S1, it would be helpful to show absolute cell number as well. If itaconate also inhibited the proliferation of cells, then that could contribute to observed effects. Replication index measurements would be useful as well.
3. Itaconate is known to activate Nrf2, but the authors do not discuss any aspect of their results from this angle, which may explain some of their results. If Nrf2 is not involved, then that could in itself be a significant finding. Performing siRNA or Cas9 mediated KO/repression of Nrf2 in naïve T cells and checking if that affects the response to itaconate could be informative.

Reviewer #2 (Remarks to the Author):

In this paper, Aso and colleagues report on how itaconate can block Th17 cell differentiation whilst promoting Treg differentiation. The mechanism appears to involve inhibition of glycolysis in Th17 cells, modulation of the SAM/SAH ratio, and inhibition of 2-HG production. They also examine chromatin accessibility and find that in response to itaconate the Th17 chromatin locus is more closed whilst that of Treg is more open. These effects of itaconate are shown to be relevant in vivo where adoptive transfer of itaconate-treated Th17 cells are inhibitory in the Th17-dependent model of EAE. These data are certainly interesting and the experiments are well carried out. I have however some questions that need to be addressed as follows.

1. The authors don't really discuss the physiological relevance of their findings. Clearly adding itaconate into their various assays is having a profound effect but are the authors suggesting cross-talk with macrophages or dendritic cells here, the likely cellular sources of the itaconate in vivo? What would happen in IRG-1-deficient mice in the EAE model that they use here? Would there be an exacerbation there and could that be shown to be due to a Treg deficit? Might it be possible to carry out a co-culture experiment with IRG1-deficient and WT macrophages which would be a source of itaconate here, and determine effects of naturally-produced itaconate on T cell differentiation? Without these kinds of experiments what we have in effect is a series of experiments with relatively high concentrations of a molecule that is definitely of interest in terms of inflammation, but without a clear picture of what the effects observed might mean for T cell differentiation in vivo.
2. How do the authors think itaconate is eliciting its effects here? Is a receptor involved and how might it signal? Or is the itaconate getting taken up by the T cells and then modulating metabolism

intracellularly? The authors should speculate on these issues but some experiments would help, for example an itaconate uptake experiment. The metabolomic analysis for example curiously doesn't measure itaconate itself. Is it substantially elevated? This could explain for example the effect on succinate.

3. The effect on 2-HG is interesting and to my knowledge this is the first time that itaconate has been shown to affect 2-HG levels. They present evidence that itaconate is an inhibitor of IDH activity. What is the likely mechanism of that inhibition, as this conclusion expands the target enzymes for itaconate.

4. Did the authors pH the itaconate to ensure that the effects aren't due to acidification?

Reviewer #3 (Remarks to the Author):

This study demonstrates a novel link between itaconate, T cell polarization and EAE. While this work is likely to be of interest to the research community some questions remain:

Major concerns/comments:

1. Itaconate is a metabolite almost exclusively produced in macrophages due to their expression of Irg1. Work from Meiser and colleagues (2018) demonstrate that itaconate can be released from macrophages but at a much lower concentration than 6 mM (the concentration applied to cells here). They report an extracellular concentration of 1-5 uM depending on the activation state of the cells. Furthermore, they report a circulating level of itaconate of just ~0.5 uM. As such I question the concentration selected here. How was it selected? Is it dose-dependent? Were concentrations lower than 3 mM tested?
2. The second major concern I have is how the authors suggest this metabolite enters T cells? Itaconate is a charged dicarboxylate and as such cannot passively enter cells. It must be a transport-mediated process. The authors do not address this/propose how these cells can sequester extracellular itaconate.
3. The authors did not show itaconate levels in T cells in their metabolomics assessment (Fig. 5). How do the levels of itaconate change in these cells post treatment?
4. Do structurally similar metabolites exert any biological effect at similar concentrations to those used here?
5. The authors show that *rorc* expression and ROR γ t is increased with itaconate, but *Il17a* and *f* expression are decreased, how do they explain this discrepancy?

6. In the EAE model (Figure 2) is the expression of other inflammatory cytokines e.g. IL-1 β and indeed the abundance of other pathogenic cell types such as neutrophils and inflammatory monocytes which are key drivers of pathology in this setting altered by itaconate?

7. The effect of itaconate treatment ex vivo on EAE is extremely impressive. Is supplementation of itaconate alone sufficient to modulate the progressive of EAE in less reductionist models e.g. MOG, CFA? This would increase the clinical relevance of the finding substantially.

8. Along similar , is itaconate supplementation after adoptive transfer of Th17-polarizing CD4+ T cells from 2D2 mice sufficient to reduce EAE severity?

Minor Concerns:

1. The language and writing needs improvement

2. Did itaconate affect OCR of Th17/Treg cells given its inhibitory effect on SDH?

3. Absence of change in HIF-1 β levels does not mean that the activity of this transcription factor is not altered. This can be determined by looking at downstream targets of HIF e.g. LDHA, VEGF, PHD3 etc, many of which are altered by itaconate at the transcriptome level. I don't believe a role for HIF1 β can be formally ruled out given the current data.

4. In Figure 3 it would be helpful to indicate which pathways are up-regulated and which are down-regulated by itaconate, this is not clear from the data presentation.

5. How is itaconate affecting the levels of distal metabolites e.g. glutamate?

6. How was the itaconate solution prepared? A solution made from itaconic acid will be extremely acidic, was this pH'd? This is not mentioned in the methods.

7. How does itaconate only affect IDH in Tregs and not Th17 cells?

Reviewer's comments:

Reviewer #1 (Remarks to the Author)

In previous studies, itaconate, a TCA cycle intermediate, was shown to have anti-inflammatory effects, acting through activation of Nrf2 and, potentially, by inhibiting succinate dehydrogenase, to attenuate cytokine production in macrophages. In this study, the authors show that exogenously administered itaconate inhibits “pathogenic” Th17 differentiation and promotes Tregs. They demonstrate that in vitro exposure to itaconate renders myelin-specific Th17-polarized T cells less pathogenic in the EAE model. They provide evidence that itaconate inhibits glycolysis and also alters the amount of SAM, that, in turn, regulates epigenetic modifications. A potentially interesting finding was the reduced methylation index (SAM/SAH ratio) in itaconate-treated Th17, but not Treg cells, while 2-HG levels were reduced in Treg, but not Th17 cells. The authors link these observations to reduced accessibility of the Il17a locus for transcription factor binding, presumably through less histone methylation, after itaconate treatment. The mechanism for the anti-inflammatory effect of itaconate has focused on macrophages, and a thorough analysis of its influence on other cells involved in inflammation, including T cells, is needed. Although this paper provides strong data as to the effect of exogenous itaconate on Th17 cell function in vitro along with a plausible explanation for how itaconate may influence the Th17:Treg balance, the relevance in vivo is not addressed, and the quality of some of the data makes it difficult to determine the likelihood of the proposed mechanism. Major points are listed below:

We thank the reviewer for the time and advice and are happy to respond to the raised questions.

1. The authors describe itaconate as an endogenous metabolite, but test its effects when added exogenously. What is the context in which T cells are exposed to exogenous itaconate rather than having their internal itaconate level modulated?

We appreciate the reviewer for raising this very important issue and for giving us the opportunity to highlight translational implications in the study. The aim of this study was to evaluate the

therapeutic potential of ITA to treat T cell-related diseases rather than assess the physiological role of this metabolite in T cells. As the reviewer pointed out, ITA is a metabolite exclusively produced by macrophages in which Irg1 is highly expressed. Supporting the fact, we identified that the expression of Irg1 was not upregulated in T cells (**new Supplementary Fig. 1f**). A previous study has shown ITA could be released from activated macrophages at an extracellular concentration of 1–5 μM (**Meiser J. et al. Oncotarget. 2018**). We speculated that ITA produced by macrophages affects T cell differentiation, but we demonstrated that the effect of ITA on Th17 and Treg differentiation was not statistically significant at a lower concentration than 1 mM (**new Supplementary Fig. 1g, h**). Altogether, these results suggested that T cell differentiation is not affected by macrophages-derived ITA *in vivo*.

Herein, we demonstrated ITA supplementation at published supraphysiological concentrations for macrophages (**Swain A. et al. Nat Metab. 2020**) inhibited Th17 cell differentiation and promoted Treg cell differentiation. We also identified that ITA treatment *in vitro* increased intracellular levels of ITA in activated naive CD4⁺ T cells after CD3/CD28 stimulation (**new Supplementary Fig. 5c**). To confirm the potential of ITA as a treatment option for autoimmune diseases, we have performed two additional experiments *in vivo*. First, we treated EAE mice with 50 mg kg⁻¹ ITA via intraperitoneal injection every other day from day 0 to day 14 post-immunization with myelin oligodendrocyte glycoprotein (MOG)₃₅₋₅₅ and complete Freund's adjuvant (CFA). We found that ITA markedly reduced the disease activity of EAE compared to vehicle treatment (**new Fig. 2a, b**). Second, the intraperitoneal injection of ITA to Rag1KO recipient mice following the adoptive transfer of Th17-polarizing CD4⁺ T cells from 2D2 mice significantly attenuated the severity of the adoptive transfer EAE model (**new Supplementary Fig. 2c, d**). These novel observations confirm the robust therapeutic potential of ITA and suggest that this metabolite could be a candidate for future therapies in autoimmune diseases. After incorporating this important suggestion, we have revised the following sentences in the main text:

In abstract: "Herein, we show that itaconate (ITA), an immunomodulatory metabolite, supplementation inhibited Th17 cell differentiation and promoted Treg cell differentiation by orchestrating the metabolic and epigenetic reprogramming."

In Introduction: "Therefore, in this study, we aimed to identify the role of ITA in regulating T cell differentiation and its potential as a candidate to treat T cell-mediated autoimmune diseases."

2. Some of the metabolic effects of itaconate revealed by the authors (such as inhibition of IDH1/2) were described previously in macrophages, and itaconate is known to be a regulator of the TCA cycle.

We thank the reviewer for highlighting this. Yes, the inhibition of SDH by ITA has been well described in macrophages (**Lampropoulou V. *et al.* Cell Metab. 2016**). We have cited the work of Lampropoulou V. *et al.* in our manuscript as follow: “Given that ITA inhibits the enzymatic activity of succinate dehydrogenase (SDH) in macrophages²⁹, we also evaluated mitochondrial OXPHOS.” In this study, we demonstrated that the ITA modified the activities of MAT and IDH1/2 and affected T cell metabolism and differentiation. To the best of our knowledge, our results is the first study to explore the effect of ITA on the modulation of these enzymes in T cells.

3. The effects of exogenous itaconate on glycolysis are well demonstrated. However, the analysis of the T cell epigenetic program is superficial, and does not provide much new insight. For example, the ChIP of ROR γ t shows very little specificity, only 2-fold over IgG, and is difficult to evaluate without other loci being included (both positive and negative loci for binding). The best negative control would be to use ROR γ t-deficient T cells. It is also difficult to judge from Fig 6b whether there are significant differences in accessibility, as only tracings of the Il17a and Foxp3 loci are shown and there are no statistical analyses based on whole genome profiles of the ATAC-seq. It is important to show other loci that have differences with itaconate, as well as characterize regions showing appearance or disappearance of new/distinct peaks. Is the GM-CSF (Csf2) locus altered? Are there chromatin accessibility changes in glycolysis genes?

We would like to thank the reviewer for the helpful comments. Unfortunately, because of the difficulties imposed by COVID-19 pandemic, we could not obtain ROR γ t-deficient mice. We are afraid we cannot make it up by availing the mice and performing the additional experiments within the stipulated review timeframe. Therefore, to address the reviewer's suggestion, we have added naïve CD4⁺ T cell as a negative control in the ChIP experiments (**new Fig. 6a**). Actually, our results showed that ITA suppressed ROR γ t binding to the Il17a promoter. To provide supporting evidence for our findings, we have cited the study of Endo *et al.* that showed

similar specificity of the anti-ROR γ t antibody for *Il17a* promoter region (Endo Y. *et al. Cell Rep.* 2015).

In addition, we have performed additional analyses based on ATAC-seq and RNA-seq datasets. As RNA expression depends on DNA accessibility, we have integrated the ATAC-seq and RNA-seq datasets (new Fig. 6b). Overall, 703 differentially expressed genes showed differential accessibility between ITA-treated and control T cells under Th17 or Treg conditions (new Fig. 6c). These genes included *Il17a*, glycolysis-related genes, and *Foxp3*, but not *Csf2*.

4. Performing RNA-seq and/or ATAC-seq and/or metabolomics with itaconate-untreated and -treated naïve T cells under Th0 conditions could provide insight into how the metabolite affects the cellular program in the absence of cytokine signals. Pre-treatment of cells with itaconate followed by polarization in Th17 or Treg conditions could also provide insight into how itaconate functions.

We appreciate the reviewer for the constructive suggestions. We have performed RNA-seq using Th0 cells with or without ITA treatment. We identified 1,461 DEGs between ITA-treated and control T cells under the Th0 condition (new Supplementary Fig. 4a). In KEGG pathway analysis, these genes presented main pathways different from those presented by Th17 and Treg RNA-seq data (new Supplementary Fig. 4b). The absence of 'glycolysis' and 'HIF-1 signaling pathway' in the overlap between DEGs between ITA-treated and control T cells under Th0, Th17, and Treg conditions suggested that ITA strongly induced these metabolic changes dependent on unique cytokine signals for Th17 or Treg differentiation (new Supplementary Fig. 4c). Furthermore, pre-treatment of Th0 cells with ITA followed by polarization in Th17 or Treg conditions induced no significant change in Th17 differentiation and a slight increase in Treg differentiation (new Supplementary Fig. 4d, e). Altogether, these findings suggest that the coordination between ITA treatment and unique cytokine signals is essential for the regulation of Th17/Treg differentiation in ITA-treated T cells.

Minor points:

1. How does treatment with itaconate compare to that with dimethyl itaconate, which has been used by other groups because it is more membrane-permeable?

We analyzed the differentiation and cell viability of murine naive CD4⁺ T cells from wild-type B6 mice activated under Th17 and Treg cell conditions with or without dimethyl ITA (DI, 0, 0.1, 0.2, and 0.4 mM) treatment. The results demonstrated that DI inhibited Th17 differentiation but did not affect Treg differentiation (**Fig. 1a, b for the Reviewer**). The concentration of DI was selected based on the viability data (**Fig. 1c, d for the Reviewer**) and published supraphysiological concentrations for macrophages (**Swain A. et al. Nat Metab. 2020**). Furthermore, Swain et al. have shown that DI was not converted to intracellular itaconate in macrophages following exogenous treatment and led to distinct immunologic impacts compared to treatment with exogenous unmodified ITA (**Swain A. et al. Nat Metab. 2020**). Therefore, we did not include these results in the manuscript.

Fig.1 for the Reviewer a–d Cumulative data of the differentiation (**a, b**) and cell viability (**c, d**) of murine naive CD4⁺ T cells from wild-type B6 mice activated under Th17 and Treg cell conditions in the presence or absence of dimethyl itaconate (DI, 0, 0.1, 0.2, and 0.4 mM) after 3 days of culture. *P* values are calculated using one-way ANOVA with Bonferroni post hoc test for (**a–d**). Data are representative of mean ± s.e.m.

2. While the authors show that cell viability is not affected in Fig S1, it would be helpful to show absolute cell number as well. If itaconate also inhibited the proliferation of cells, then that could contribute to observed effects. Replication index measurements would be useful as well.

We appreciate the reviewer for this insightful suggestion. As suggested, we have added absolute cell numbers in **Supplementary Fig. 1c** and performed additional experiments demonstrating that the proliferation of T cells was not affected by ITA *in vitro* (**Supplementary Fig. 1d**).

3. Itaconate is known to activate Nrf2, but the authors do not discuss any aspect of their results from this angle, which may explain some of their results. If Nrf2 is not involved, then that could in itself be a significant finding. Performing siRNA or Cas9 mediated KO/repression of Nrf2 in naïve T cells and checking if that affects the response to itaconate could be informative.

We thank the reviewer for the attentiveness and have demonstrated that ITA inhibited Th17 cell differentiation and promoted Treg cell differentiation in Nrf2-knockout T cells (**new Fig. 5h, i**). These data suggest that Nrf2 activation is unlikely to be the main mechanism of the regulation of Th17/Treg cell differentiation by ITA.

Reviewer #2 (Remarks to the Author)

In this paper, Aso and colleagues report on how itaconate can block Th17 cell differentiation whilst promoting Treg differentiation. The mechanism appears to involve inhibition of glycolysis in Th17 cells, modulation of the SAM/SAH ratio, and inhibition of 2-HG production. They also examine chromatin accessibility and find that in response to itaconate the Th17 chromatin locus is more closed whilst that of Treg is more open. These effects of itaconate are shown to be relevant *in vivo* where adoptive transfer of itaconate-treated Th17 cells are inhibitory in the Th17-dependent model of EAE. These data are certainly interesting and the experiments are well carried out. I have however some questions that need to be addressed as follows.

We would like to thank the reviewer for her/his time and helpful feedback. We are pleased to read that the reviewer finds the paper interesting and are happy to address the questions raised.

1. The authors don't really discuss the physiological relevance of their findings. Clearly adding itaconate into their various assays is having a profound effect but are the authors suggesting cross-talk with macrophages or dendritic cells here, the likely cellular sources of the itaconate *in vivo*? What would happen in IRG-1-deficient mice in the EAE model that they use here? Would there be an exacerbation there and could that be shown to be due to a Treg deficit? Might it be possible to carry out a co-culture experiment with IRG1-deficient and WT macrophages which would be a source of itaconate here, and determine effects of naturally-produced itaconate on T cell differentiation? Without these kinds of experiments what we have in effect is a series of experiments with relatively high concentrations of a molecule that is definitely of interest in terms of inflammation, but without a clear picture of what the effects observed might mean for T cell differentiation *in vivo*.

We thank the reviewer for raising this very important point and for allowing us to highlight the translational implications of the study. The aim of this study was to evaluate the therapeutic potential of ITA to treat T cell-related diseases rather than assess the physiological role of this metabolite in T cells. As the reviewer pointed out, we agree that addressing the suggested information will strengthen our results. Unfortunately, we have faced difficulties obtaining IRG-1-deficient mice due to COVID-19 pandemic and they will not be available in the timeframe of the review process. Therefore, we have sought another approach to address the reviewer's suggestion. ITA is a metabolite produced by macrophages in which *Irg1* is highly expressed. In contrast, we demonstrated that the expression of *Irg1* was not upregulated in T cells (**new Supplementary Fig. 1f**). Moreover, a previous study has shown ITA could be released from activated macrophages at an extracellular concentration of 1–5 μ M (**Meiser J. et al. Oncotarget. 2018**). However, here, we demonstrated that the effect of ITA on Th17 and Treg differentiation was not statistically significant at a lower concentration than 1 mM (**new Supplementary Fig. 1g, h**). Altogether, these results suggested that T cell differentiation is not affected by macrophages-derived ITA *in vivo*.

To confirm the potential of ITA as a treatment option for autoimmune diseases, we have performed two additional experiments *in vivo*. First, we treated EAE mice with 50 mg kg⁻¹ ITA via intraperitoneal injection every other day from day 0 to day 14 post-immunization with myelin oligodendrocyte glycoprotein (MOG)₃₅₋₅₅ and complete Freund's adjuvant (CFA). We found that ITA markedly reduced the disease activity of EAE compared to vehicle treatment (**new Fig. 2a,**

b). Second, the intraperitoneal injection of ITA to Rag1KO recipient mice following the adoptive transfer of Th17-polarizing CD4⁺ T cells from 2D2 mice significantly attenuated the severity of the adoptive transfer EAE model (**new Supplementary Fig. 2c, d**). These novel observations confirm the robust therapeutic potential of ITA and suggest that this metabolite could be a candidate for future therapies in autoimmune diseases.

After incorporating this important suggestion, we have revised the following sentences in the main text:

In abstract: "Herein, we show that itaconate (ITA), an immunomodulatory metabolite, supplementation inhibited Th17 cell differentiation and promoted Treg cell differentiation by orchestrating the metabolic and epigenetic reprogramming."

In Introduction: "Therefore, in this study, we aimed to identify the role of ITA in regulating T cell differentiation and its potential as a candidate to treat T cell-mediated autoimmune diseases."

2. How do the authors think itaconate is eliciting its effects here? Is a receptor involved and how might it signal? Or is the itaconate getting taken up by the T cells and then modulating metabolism intracellularly? The authors should speculate on these issues but some experiments would help, for example an itaconate uptake experiment. The metabolomic analysis for example curiously doesn't measure itaconate itself. Is it substantially elevated? This could explain for example the effect on succinate.

We thank the reviewer for this insightful suggestion. As suggested by the reviewer, we performed additional experiments and demonstrated that ITA treatment *in vitro* increased intracellular levels of ITA in activated naive CD4⁺ T cells after CD3/CD28 stimulation (**new Supplementary Fig. 5c**). Further, the expression of *Irg1*, which encodes a mitochondrial enzyme catalyzing the production of ITA, was not upregulated in T cells (**new Supplementary Fig. 1f**). These data suggest that exogenous ITA was taken up by the T cells as well as macrophages (**Swain A. et al. Nat Metab. 2020**) and modulated metabolism intracellularly.

3. The effect on 2-HG is interesting and to my knowledge this is the first time that itaconate has been shown to affect 2-HG levels. They present evidence that itaconate is an inhibitor of IDH activity. What is the likely mechanism of that inhibition, as this conclusion expands the target enzymes for itaconate.

We thank the reviewer for highlighting this point. A previous study has shown that itaconate binds directly to TET-family DNA dioxygenases like the co-substrate α -ketoglutarate and inhibits its catalytic activity (**Chen LL. *et al.* Nat Cell Biol. 2022**). Since isocitrate, a co-substrate of IDH, has a similar structure to itaconate, we speculated that ITA directly inhibited IDH activity. As shown in **new Fig. 5f, f**, our results supported this speculation and showed that ITA directly inhibited the activity of purified wild-type IDH1 and 2. Further, we prepared whole-cell extract including IDH from Th17- and Treg-polarizing T cells and demonstrated that itaconate inhibits the IDH activity of the extract derived from Treg, but not Th17 (**new Supplementary Fig. 5f**). Furthermore, our metabolomics data revealed that ITA decreased isocitrate levels in Treg, but not Th17 (**Fig. 5a**). Based on these data, we inferred that the difference in the co-substrate levels of each cell may affect the inhibitory effect of ITA in IDH enzyme-catalyzed reaction. This interpretation has been included in the manuscript.

4. Did the authors pH the itaconate to ensure that the effects aren't due to acidification?

We thank the reviewer for this useful comment. The medium, including ITA, was adjusted to pH 7.4 with 1N NaOH at 37°C. We have included this sentence in the Methods section of the manuscript.

Reviewer #3 (Remarks to the Author)

This study demonstrates a novel link between itaconate, T cell polarization and EAE. While this work is likely to be of interest to the research community some questions remain:

We would like to thank the reviewer for the suggestions and hope they are now sufficiently addressed.

Major concerns/comments:

1. Itaconate is a metabolite almost exclusively produced in macrophages due to their expression of Irg1. Work from Meiser and colleagues (2018) demonstrate that itaconate can be released from macrophages but at a much lower concentration than 6 mM (the concentration applied to cells here). They report an extracellular concentration of 1-5 uM depending on the activation state of the cells. Furthermore, they report a circulating level

of itaconate of just ~0.5 μ M. As such I question the concentration selected here. How was it selected? Is it dose-dependent? Were concentrations lower than 3 mM tested?

We thank the reviewer for raising this very important point. As the reviewer pointed out, ITA is a metabolite produced by macrophages in which *Irg1* is highly expressed. We have demonstrated no upregulation of *Irg1* in T cells (**new Supplementary Fig. 1f**). Moreover, we have performed additional experiments demonstrating that the effect of ITA on Th17 and Treg differentiation was not statistically significant at a lower concentration than 1 mM (**new Supplementary Fig. 1g, h**). These results suggested that T cell differentiation is not affected by macrophages-derived itaconate in vivo.

We are aware that the aim of this study is to evaluate the therapeutic effect of itaconate for T cell-mediated diseases rather than assess the physiological role of this metabolite in T cells. Given that our novel observations and a robust therapeutic potential of itaconate, it is likely that this metabolite could be considered as a candidate to treat autoimmune diseases. We selected the treatment concentrations of ITA, based on viability assessment (**Supplementary Fig. 1b**), immunological characterization, and published supraphysiological concentrations for macrophages (**Swain A. et al. Nat Metab. 2020**).

2. The second major concern I have is how the authors suggest this metabolite enters T cells? Itaconate is a charged dicarboxylate and as such cannot passively enter cells. It must be a transport-mediated process. The authors do not address this/propose how these cells can sequester extracellular itaconate.

3. The authors did not show itaconate levels in T cells in their metabolomics assessment (Fig. 5). How do the levels of itaconate change in these cells post treatment?

We appreciate the reviewer for pointing this out. We have performed additional experiments as suggested by the reviewer. ITA treatment in vitro increased intracellular levels of itaconate in activated naïve CD4⁺ T cells (**new Supplementary Fig. 5c**). Further, the expression of *Irg1*, which encodes a mitochondrial enzyme catalyzing the production of ITA, was not upregulated in T cells (**new Supplementary Fig. 1f**). These data suggested that exogenous ITA was taken up by the T cells as well as macrophages (**Swain A. et al. Nat Metab. 2020**). The high polarity and low electrophilicity of ITA result in its low cell-permeability (**Lin J. et al. Front Chem. 2021**). We agree that a transport-mediated process such as mitochondrial oxoglutarate, dicarboxylate, and

citrate carriers (Mills EL. *et al. Nature. 2018*) might be involved in the ITA uptake in T cells. We have discussed this possible mechanism in the limitations part.

4. Do structurally similar metabolites exert any biological effect at similar concentrations to those used here?

We thank the reviewer for this useful suggestion. We have selected succinate and malonate as structurally similar metabolites and found these metabolites did not affect the differentiation of Th17 and Treg cells at a concentration similar to ITA (Fig. 2a–d for the Reviewer).

Fig. 2 for the Reviewer a–d Cumulative data of the differentiation of murine naive CD4⁺ T cells from wild-type B6 mice activated under Th17 and Treg cell conditions in the presence or absence of succinate (a, b) and malonate (c, d) after 3 days of culture. P values are calculated using one-way ANOVA with Bonferroni post hoc test for (a–d). Data are representative of mean ± s.e.m.

5. The authors show that rorc expression and RORγt is increased with itaconate, but Il17a and f expression are decreased, how do they explain this discrepancy?

We previously showed in Fig. 6a that ITA suppressed RORγt binding to the *Il17a* promoter. In addition, ATAC-seq revealed that ITA leads the chromatin accessibility to closed in *Il17a* and *Il17f* loci for key transcription factors (new Fig. 6b–e). As described in the Discussion section,

we highlighted that ITA-mediated metabolic reprogramming, including reduced SAM, could change epigenetic status in *Il17a* and *Il17f* loci. Further supporting our conclusion, Th17 cells cultured under methionine restriction, which induces SAM reduction, showed reduced IL-17 production but stable ROR γ t expression (Roy DG. *et al. Cell Metab.* 2020).

6. In the EAE model (Figure 2) is the expression of other inflammatory cytokines e.g. IL-1 β and indeed the abundance of other pathogenic cell types such as neutrophils and inflammatory monocytes which are key drivers of pathology in this setting altered by itaconate?

We appreciate the reviewer's suggestion. Accordingly, we have evaluated the infiltrating neutrophils, inflammatory monocytes, and macrophages in the spinal cord of the adoptive transfer EAE model (new Supplementary Fig. 2e). The absolute number of infiltrating these cells or IL-1 β -producing cells in recipient mice administered ITA-treated cells were not significantly different from that of their counterparts (new Fig. 2i, j). These data suggested that *in vivo* induction of neutrophils, inflammatory monocytes, and macrophages is unlikely to be the main pathogenicity of the transfer EAE attenuation by ITA-treated Th17 cells.

7. The effect of itaconate treatment ex vivo on EAE is extremely impressive. Is supplementation of itaconate alone sufficient to modulate the progressive of EAE in less reductionist models e.g. MOG, CFA? This would increase the clinical relevance of the finding substantially.

As suggested by the reviewer, we have treated EAE mice with 50 mg kg⁻¹ ITA intraperitoneal injection every other day from day 0 to day 14 post-immunization with myelin oligodendrocyte glycoprotein (MOG)₃₅₋₅₅ and complete Freund's adjuvant (CFA). We found that ITA markedly reduced the disease activity of EAE compared to vehicle treatment (new Fig. 2a, b).

8. Along similar, is itaconate supplementation after adoptive transfer of Th17-polarizing CD4+ T cells from 2D2 mice sufficient to reduce EAE severity?

We appreciate the reviewer's suggestion. Accordingly, we have performed an additional experiment and found a significant reduction in clinical scores and loss of body weight in

transfer EAE model by ITA supplementation after adoptive transfer (**new Supplementary Fig. 2c, d**).

Minor Concerns:

1. The language and writing needs improvement

We apologize for the inconvenience caused by our writing. We have requested native speakers of English to proofread our manuscript.

2. Did itaconate affect OCR of Th17/Treg cells given its inhibitory effect on SDH?

We appreciate the suggestion of the reviewer and have performed additional experiments. The mitochondrial oxygen consumption rate (OCR) showed decreased basal and maximal respiration and ATP production in ITA-treated Th17- and Treg-polarizing cells. These data were added in **new Fig. 4e and f**.

3. Absence of change in HIF-1 α levels does not mean that the activity of this transcription factor is not altered. This can be determined by looking at downstream targets of HIF e.g. LDHA, VEGF, PHD3 etc, many of which are altered by itaconate at the transcriptome level. I don't believe a role for HIF1 α can be formally ruled out given the current data.

We agree with the reviewer that our data cannot exclude the possibility that the HIF-1 α pathway is affected by ITA treatment. We apologize for this exaggerated expression. According to the reviewer's comment, we have now corrected the following sentences in our revised manuscript: In abstract: "Mechanistically, ITA suppressed glycolysis and OXPHOS in Th17- and Treg-polarizing T cells."

In results: "Itaconate inhibits glycolysis under Th17- and Treg-polarizing conditions."

4. In Figure 3 it would be helpful to indicate which pathways are up-regulated and which are down-regulated by itaconate, this is not clear from the data presentation.

We agree with the reviewer and have revised the data accordingly (**new Fig. 3e and new Supplementary Fig. 3a, b**). Among Th17 signature genes, the relative expression of genes that encode the effector cytokine IL-17A (*Il17a*) and F (*Il17f*) and signal transducer and activator of transcription 3 (*Stat3*) were downregulated by ITA under Th17 condition compared to control (**new Fig. 3c and new Supplementary Fig. 3c**). Interestingly, the relative expression of gene encoding important transcription factor for Th17 polarization, including ROR γ t (*Rorc*) was not downregulated by ITA (**Supplementary Fig. 3c**). Supporting these results, Th17 cells cultured under methionine restriction, which induces SAM reduction, exhibited a similar trend towards the change of Th17 signature genes (**Roy DG. et al. Cell Metab. 2020**), which were observed in ITA-treated Th17 cells (**new Supplementary Fig. 3c**).

5. How is itaconate affecting the levels of distal metabolites e.g. glutamate?

Although we speculated that ITA inhibits glutaminase (GLS) activity, which converts glutamine to glutamate, ITA did not inhibit the enzymatic activity (**Fig. 3 for the Reviewer**). Reportedly, ITA exerts anti-inflammatory effects via activation of Nrf2 in macrophages (**Mills EL. et al. Nature. 2018**). Nrf2 is also known to facilitate glutaminolysis and redirect glutamate into anabolic pathways (**Mitsuishi Y. et al. Cancer Cell. 2012**). Therefore, we have assessed whether Nrf2 contributes to the itaconate regulation of Th17/Treg cell differentiation using Nrf2-deficient mice. The results demonstrated that ITA inhibited Th17 cell differentiation and promoted Treg cell differentiation in Nrf2-deficient T cells (**new Fig. 5h, i**), suggesting that Nrf2 activation is unlikely to be the main mechanism of the regulation of Th17/Treg cell differentiation by ITA. We are currently investigating the alteration of these distal metabolites; however, we think discussing these data is beyond the scope of this study and will extend and diffuse the currently focused discussion.

Fig. 3 for the Reviewer Enzymatic activity of glutaminase (GLS) in Th17- and Treg-polarizing T cells with or without ITA. GLS enzymes were purified from Th17- and Treg-polarizing T cells with or without ITA on day2, and the activities were assessed by GLS activity assay kit

(Biovision) according to the manufacturer's instructions. *P* values are calculated using two-tailed unpaired Student's *t*-test. Data are representative of mean \pm s.e.m.

6. How was the itaconate solution prepared? A solution made from itaconic acid will be extremely acidic, was this pH'd? This is not mentioned in the methods.

We thank the reviewer for this useful comment. The medium, including itaconate, was adjusted to pH 7.4 with 1N NaOH at 37°C. We updated this information in the revised manuscript.

7. How does itaconate only affect IDH in Tregs and not Th17 cells?

We appreciate the reviewer for highlighting this point. After performing additional experiments, we show that ITA directly inhibited the activity of IDH1 and 2 (**new Fig. 5f, g**). We prepared whole-cell extract including IDH from Th17- and Treg-polarizing T cells, respectively, and demonstrated that ITA inhibits the IDH activity of the extract derived from Treg but not from Th17 (**new Supplementary Fig. 5f**). Furthermore, our metabolomics data revealed that ITA decreased isocitrate, a co-substrate of IDH, levels in Treg, but not Th17 (**Fig. 5a**). Based on these data, we inferred that the difference in the co-substrate levels of each cell may affect the inhibitory effect of ITA in IDH enzyme-catalyzed reaction. This interpretation has been included in the manuscript.

REVIEWER COMMENTS

Reviewer #1 (Remarks to the Author):

In their revised manuscript and their rebuttal, Aso et al. have made a good effort responding to the reviewers' comments. They clarified that the purpose of their study on itaconate influence on the Th17-Treg balance is to raise the prospect of its therapeutic use in autoimmune disease. They now show that the amount of ITA produced by macrophages, and, presumably, the in vivo concentration, is far lower than the concentration needed to influence T cell differentiation, and hence is unlikely to reflect a physiological function of ITA. In new experiments, they show that administration of ITA to mice with MOG-induced EAE reduces the severity of the disease, as does ITA injection in mice receiving in vitro Th17-polarized transgenic 2D2 T cells, thus supporting their proposal of ITA as a therapeutic agent. The results are consistent with their previous demonstration of reduced pathogenicity of in vitro differentiated 2D2 Th17 cells that had been treated with ITA before adoptive transfer. However, it is unclear how much of the attenuated disease in these EAE models is due to an in vivo effect of ITA on the MOG-specific T cells rather than other cell types.

The authors also show that the mechanism by which ITA acts in vitro on differentiating T cells is independent of Nrf2, although a precise mechanism for the activity is not shown. They infer that the mechanism is by way of changes in histone methylation, based on altered SAM/SAH ratio, and differential effect on IDH1/2 in in vitro differentiated Th17 vs Treg, but this argument is based on correlation. ITA clearly alters chromatin accessibility, as shown by ATAC-seq data, and, not surprisingly, the transcriptional profiles affected by ITA are influenced by the cytokine milieu. But is there no effect on Rorc accessibility, but rather on targets of the RORgt protein, such as Il17a/f? How would this specificity be explained?

The manuscript is improved by the newly-included experiments, and the data suggest that ITA can indeed alter the course of EAE in vivo. Whether itaconate ameliorates EAE solely by inhibiting the Th17 cell effector program and/or the Th17/Treg ratio remains unclear, and the precise mechanism by which ITA affects the in vitro differentiation of Th17 and Treg cells is also not yet explained by the current study. But a better understanding of how ITA differentially targets the Th17 and Treg transcriptional programs could indeed suggest a path towards use of metabolism reprogramming drugs to treat inflammatory disease.

Reviewer #2 (Remarks to the Author):

The authors have done an excellent job addressing the concerns I raised and I'm happy to recommend acceptance.

Reviewer #3 (Remarks to the Author):

The authors have addressed most of my concerns and I think the manuscript is improved however I think it needs to be explicitly stated that while these data support a therapeutic role for itaconate in EAE, due to the concentrations used that a T cell will never experience physiologically, they do not support a physiological role for itaconate as a regulator for T cells. At the doses found in the circulation/secreted from macrophages there was no effect observed. This needs to be clearly stated and I believe the title should also be revised to reflect this.

Reviewer's comments:

Reviewer #1 (Remarks to the Author)

In their revised manuscript and their rebuttal, Aso et al. have made a good effort responding to the reviewers' comments. They clarified that the purpose of their study on itaconate influence on the Th17-Treg balance is to raise the prospect of its therapeutic use in autoimmune disease. They now show that the amount of ITA produced by macrophages, and, presumably, the in vivo concentration, is far lower than the concentration needed to influence T cell differentiation, and hence is unlikely to reflect a physiological function of ITA. In new experiments, they show that administration of ITA to mice with MOG-induced EAE reduces the severity of the disease, as does ITA injection in mice receiving in vitro Th17-polarized transgenic 2D2 T cells, thus supporting their proposal of ITA as a therapeutic agent. The results are consistent with their previous demonstration of reduced pathogenicity of in vitro differentiated 2D2 Th17 cells that had been treated with ITA before adoptive transfer. However, it is unclear how much of the attenuated disease in these EAE models is due to an in vivo effect of ITA on the MOG-specific T cells rather than other cell types.

First of all, we thank the reviewer for the appreciation. Once again, thank you for your insightful comments, which we believe have strengthened the presentation of our work. We are glad that our responses and the revisions made in the previous round of revision were satisfactory. As for the point highlighted here, we agree with the limitation of the MOG-induced EAE model and the possible effects of ITA on cell types other than T cells, such as anti-inflammatory effects on macrophages (Swain A. *et al. Nat Metab.* 2020). We have mentioned this limitation in the section of EAE.

In “*Itaconate ameliorated the EAE model*” section: The MOG-induced EAE model could not strictly exclude the possible effects of ITA on cell types other than T cells, such as the anti-inflammatory effect on macrophages.

However, we recognize the attenuation of the transfer EAE model, which is a more T cell-dependent model, by ITA-treated Th17 cells (Fig. 2c, d) and that ITA supplementation (Supplementary Fig. 2c, d) complements the limitation. To evaluate the effect of ITA on

pathogenic cell types other than T cells in transfer EAE, we have evaluated the infiltrating neutrophils, inflammatory monocytes, and macrophages, which drive T cells that mediate pathology of EAE (McGinley AM. *et al. Immunity. 2020*), in the spinal cord of the adoptive transfer EAE model (Supplementary Fig. 2e). The absolute number of these infiltrated cells or IL-1 β -producing cells in recipient mice administered ITA-treated cells was not significantly different from that in their counterparts (Fig. 2i, j). These data suggest that *in vivo* induction of neutrophils, inflammatory monocytes, and macrophages is unlikely to be the main pathogenicity of the attenuation of transfer EAE by ITA.

The authors also show that the mechanism by which ITA acts *in vitro* on differentiating T cells is independent of Nrf2, although a precise mechanism for the activity is not shown. They infer that the mechanism is by way of changes in histone methylation, based on altered SAM/SAH ratio, and differential effect on IDH1/2 in *in vitro* differentiated Th17 vs Treg, but this argument is based on correlation.

We thank the reviewer for this comment. We quantified the expression of Nrf2 (*Nfe2l2*) and *Hmox1*, a prototypical Nrf2 target gene, in Th17- and Treg-polarizing T cells with or without ITA. ITA showed a trend toward increased gene expression of *Nfe2l2*; however, the expression of the downstream target gene transcripts was not significantly increased (new Supplementary Fig. 5i). Immunoblotting exhibited that the protein levels of NRF2 and heme oxygenase 1 (HMOX1) did not increase in ITA treated Th17- and Treg- polarizing T cells compared to those in the control (new Supplementary Fig. 5j–l). Therefore, we inferred that ITA did not activate Nrf2 and its downstream pathway in Th17- and Treg-polarizing T cells. These data complement the result shown in Fig. 5h, i, and highlight that the Nrf2 pathway is not the main mechanism of the regulation of Th17/Treg cell differentiation by ITA.

To address the reviewer's concern, we examined whether increasing intracellular SAM or 2-HG restored the effect of ITA on Th17 or Treg cell differentiation, respectively. Intracellular SAM is derived from extracellular methionine in Th17-polarizing T cells (Roy DG. *et al. Cell Metab. 2020*). Treatment with escalating doses of methionine gradually promoted Th17 differentiation in ITA-treated Th17-polarizing T cells (new Supplementary Fig. 5g). Additionally, increasing doses of cell-permeable 2-HG (Lv H. *et al. Cell Metab. 2021*) impaired Treg differentiation in ITA-treated Treg-polarizing T cells (new Supplementary Fig. 5h). These data indicate that the SAM or 2-HG levels affect the polarization programs of ITA-treated Th17 or Treg, respectively. However, as

multiple pathways are involved in the anti-inflammatory effect of ITA on macrophages (Bambouskova M. *et al. Nature. 2018* and Mills EL. *et al. Nature. 2018*), unknown mechanisms may contribute to the program underlying ITA-regulated T cell differentiation. We agree that our proposed mechanism may not solely elucidate the regulation of T cell differentiation by ITA. In response to the reviewer's comment, we have discussed this possibility in the limitations section.

In the limitations section: As multiple pathways are involved in the anti-inflammatory effect of ITA on macrophages, unknown mechanisms may contribute to the regulation of ITA-mediated T cell differentiation.

ITA clearly alters chromatin accessibility, as shown by ATAC-seq data, and, not surprisingly, the transcriptional profiles affected by ITA are influenced by the cytokine milieu. But is there no effect on Rorc accessibility, but rather on targets of the RORgt protein, such as Il17a/f? How would this specificity be explained?

We appreciate the reviewer for this comment. As suggested by the reviewer, we focused on the discrepancy that ROR γ t expression was increased with ITA, but IL-17A expression was decreased. Concordant with our study, a previous study has reported that Th17 cells cultured under methionine restriction, which induces SAM reduction, showed reduced IL-17 production but stable ROR γ t expression (Roy DG. *et al. Cell Metab. 2020*). Decreased SAM levels induced demethylation of histone H3K4 trimethylation (H3K4me3) at the *Il17a* promoter, resulting in the downregulation of *Il17a* gene expression in Th17 cells. In the same study, SAM reduction also resulted in the demethylation of H3Kme3 around the *Rorc* transcription start site, but this did not decrease *Rorc* gene expression. Thus, the influence of SAM reduction on the histone modification may not be specific to *Il17a* gene loci, and additional mechanisms may affect *Rorc* expression. Further study are required to elucidate this selectivity of gene expression. We have discussed these issues in the limitations section.

In the limitations section: We focused on the discrepancy that ROR γ t expression was increased with ITA, but IL-17A expression was decreased. Concordant with our study, a previous study has reported that Th17 cells cultured under methionine restriction, which induces SAM reduction, showed reduced IL-17 production but stable ROR γ t expression. Decreased SAM levels induced demethylation of histone H3K4me3 at the *Il17a* promoter, resulting in the downregulation of *Il17a* gene expression in Th17 cells. In the same study, SAM reduction also resulted in the demethylation of H3Kme3 around the *Rorc* transcription start site,

but this did not decrease *Rorc* gene expression. Thus, the influence of SAM reduction on the histone modification may not be specific to *Il17a* gene loci, and additional mechanisms may affect *Rorc* expression. Further studies are required to reveal this selectivity of gene expression.

Reportedly, ROR γ t possesses some binding sites other than the promoter region of *Il17a* (Wang X. *et al. Immunity*. 2012). Moreover, the promoter region is marked by H3K4me3 (Tripathi SK. *et al. Immunol Rev*. 2014), which is the most labile mark affected by the restriction of the methionine cycle (Roy DG. *et al. Cell Metab*. 2020). Therefore, we focused on the promoter region, which mainly regulates *Il17a* expression (Ivanov II. *et al. Cell*. 2006; Kurebayashi Y. *et al. Genes Cells*. 2013). In Fig. 6a, we showed that ITA suppressed ROR γ t binding to the *Il17a* promoter. In motif analysis of ATAC-seq data, ITA decreased chromatin accessibility at binding sites of key transcription factors, which bind to the promoter regions of *Il17a* locus cooperatively with ROR γ t (Fig. 6e), supporting the results shown in Fig. 6a. We speculated that ITA might not change the accessibility of ROR γ t at *Il17a* locus other than at the promoter region, and the accessibility in ROR γ t-binding sites does not appear to change. We have discussed this possibility in the limitations section.

In the limitations section: In the motif analysis, ROR γ t was not detected in the Th17 group. Reportedly, ROR γ t possesses some binding sites other than the promoter region of *Il17a*, and the promoter region is marked by H3K4me3, the most labile mark affected by the restriction of the methionine cycle. Therefore, we speculated that ITA might not change the accessibility of ROR γ t at the *Il17a* locus other than at the promoter region, and the accessibility in these ROR γ t-binding sites does not appear to change.

The manuscript is improved by the newly-included experiments, and the data suggest that ITA can indeed alter the course of EAE in vivo. Whether itaconate ameliorates EAE solely by inhibiting the Th17 cell effector program and/or the Th17/Treg ratio remains unclear, and the precise mechanism by which ITA affects the in vitro differentiation of Th17 and Treg cells is also not yet explained by the current study. But a better understanding of how ITA differentially targets the Th17 and Treg transcriptional programs could indeed suggest a path towards use of metabolism reprogramming drugs to treat inflammatory disease.

We would like to thank the reviewer again for their time and helpful feedback. We are grateful for all constructive suggestions. We hope that the changes made in the manuscript are satisfactory and that our revised manuscript now addresses the concerns by the reviewer.

Reviewer #2 (Remarks to the Author):

The authors have done an excellent job addressing the concerns I raised and I'm happy to recommend acceptance.

We thank Reviewer #2 again for acknowledging the novelty and significance of the study and providing constructive criticisms. Their suggestions were extremely helpful in improving the quality of the manuscript. Thank you again for your positive feedback and recommendation for the acceptance of our manuscript.

Reviewer #3 (Remarks to the Author):

The authors have addressed most of my concerns and I think the manuscript is improved however I think it needs to be explicitly stated that while these data support a therapeutic role for itaconate in EAE, due to the concentrations used that a T cell will never experience physiologically, they do not support a physiological role for itaconate as a regulator for T cells. At the doses found in the circulation/secreted from macrophages there was no effect observed. This needs to be clearly stated and I believe the title should also be revised to reflect this.

We thank Reviewer #3 for their constructive suggestions in the previous round of revision. We are glad that the revisions and responses are found to be satisfactory. We thank the reviewer for this helpful suggestion. After incorporating this important suggestion, we have revised the following sentences in the main text:

In title: "Itaconate ameliorates autoimmunity by modulating T cell imbalance via metabolic and epigenetic reprogramming."

In “*Itaconate modulates Th17 and Treg cell differentiation*” section: “Altogether, these results suggested that T cell differentiation is not affected by macrophages-derived ITA *in vivo*.”

Once again, we thank all the reviewers for their helpful comments and suggestions. We hope that the additional revisions have satisfactorily addressed the reviewers' concerns and improved the overall quality of the study.

REVIEWERS' COMMENTS

Reviewer #1 (Remarks to the Author):

The authors have addressed my concerns and the manuscript is improved by inclusion of the potential caveats.

Reviewer #3 (Remarks to the Author):

This paper is now acceptable